# Different roles of D1/D2 medium spiny neurons in the nucleus accumbens in pair bond formation of male mandarin voles

**Lizi Zhang[1†], Yishan Qu[1†], Larry J Young[2,3], Wenjuan Hou[1], Limin Liu[1], Jing Liu[1], Yuqian Wang[1], Lu Li[1], Xing Guo[1], Yin Li[1], Caihong Huang[1], Zijian Lv[1], Yi-Tong Li[1], Rui Jia[1], Ting Lian[4], Hao Feng[1], Hui Qiao[1], Zhixiong He[1]\*, Fa-Dao Tai[1]\***

[1]Institute of Brain and Behavioural Sciences, College of Life Sciences, Shaanxi Normal University, Xi'an, China; [2]Department of Psychiatry and Behavioral Sciences, Silvio O. Conte Center for Oxytocin and Social Cognition, Center for Translational Social Neuroscience, Emory National Primate Research Center, Emory University, Atlanta, United States; [3]Center for Social Neural Networks, University of Tsukuba, Tsukuba, Japan; [4]Research Center for Prevention and Treatment of Respiratory Disease, School of Clinical Medicine, Xi'an Medical University, Xi'an, China

**\*For correspondence:**
hezhixiong@snnu.edu.cn (ZH);
taifadao@snnu.edu.cn (F-DT)

[†]These authors contributed equally to this work

## eLife Assessment

This **important** study advances our understanding of the role of dopamine in modulating pair bonding in mandarin voles by examining dopamine signaling within the nucleus accumbens across various social stimuli using state-of-the-art causal perturbations. The evidence supporting the findings is **compelling**, particularly cutting-edge approaches for measuring dopamine release as well as the activity of dopamine receptor populations during social bonding. Some concerns remain about the statistical analyses.

**Abstract** The mesolimbic dopamine (DA) system has been implicated in pair bond formation. However, the involvements of DA release, real-time activities, and electrophysiological activities of D1/D2 medium spiny neurons (MSNs) in the nucleus accumbens (NAc) shell in pair bonding remain unclear. This work verified that male mandarin voles after pair bonding released higher levels of DA in the NAc shell and displayed higher levels of D1 MSNs activity and lower levels of D2 MSNs activity upon sniffing their partners compared to upon sniffing an unknown female. Moreover, pair bonding induced differential alterations in both synaptic plasticity and neuronal intrinsic excitability in both D1 MSNs and D2 MSNs. In addition, chemogenetic inhibition of ventral pallidum (VP)-projecting D2 MSNs in the NAc shell enhanced pair bond formation, while chemogenetic activation of VP-projecting D2 MSNs in the NAc shell inhibited pair bond formation. These findings suggest that different neuronal activity of NAc shell D1 MSNs / D2 MSNs regulated by increasing DA release after pair bonding may be a neurobiological mechanism underlying pair bond formation.

## Introduction

In monogamous animals, pair bonding is a selective, lasting relationship between two adult individuals forming a breeding pair (*Bales et al., 2021*; *López-Gutiérrez et al., 2021*). Pair bonding is promoted by enduring social attraction between partners, and after sexual activity, partners remain together and jointly defend resources (*Carter and Perkeybile, 2018*; *Numan and Young, 2016*). In monogamous

**eLife digest** In some animals, including humans, breeding pairs form strong, lasting bonds. These connections can be useful when raising offspring, especially in species where rearing takes a long time. Chemicals in the brain play a part in forming these bonds, including dopamine, a chemical involved in triggering feelings of pleasure and reward.

The nucleus accumbens is a region of the brain that has roles in decision-making, motivation and reward processing. It is the centre that connects the motivation to do something with taking the action, and it has a fundamental role in partner selection and pair bonding. Dopamine is an important chemical in the nucleus accumbens, and two types of neurons in this region of the brain, known as D1 and D2 spiny neurons, each carry a different dopamine receptor. Changes in dopamine release, or changes in how the D1 and D2 neurons in the nucleus accumbens respond to this chemical, likely shape how pair bonding occurs.

To learn more, Zhang, Qu et al. examined dopamine levels and neuron activity in the nucleus accumbens of mandarin voles, a type of rodent that forms monogamous relationships. The researchers found that after the voles paired up, their brains released higher levels of dopamine in the nucleus accumbens, triggering feelings of pleasure. Additionally, Zhang, Qu et al. found that when the voles sniffed their partners, the activity of D1 neurons increased, while that of D2 neurons decreased. Forming a bond changed how the different neurons connected and responded to dopamine, indicating that each type of neuron plays a different role in bond formation.

These findings help us understand how pair bonding occurs, but the results could also provide insights into how other types of close connections are established and maintained. This information could also help researchers better understand the neurological basis of neurodevelopmental conditions or mental health issues, such as social anxiety.

animals, pair bonding is usually characterized by partner preference (*Carp et al., 2016*), which is defined as the selective social preference for a pair mate rather than a stranger. In addition, both male and female prairie voles develop partner preferences following mating, and common neural pathways are engaged during pair bonding (*Numan and Young, 2016*). This relationship provides a secure psychological base for both partners and acts as a social buffer against life's stresses in humans. Research showed that married people—especially in stable attachments—live longer than unmarried people (*Jia and Lubetkin, 2020*; *Rendall et al., 2011*). In addition, the intimacy of couples affects not only their psychological status (e.g. by reducing instances of depression) and the rate of disease progression but also their immune system and cardiovascular health (*Ford and Young, 2021a*; *Frasure-Smith et al., 2000*; *Leserman et al., 1999*; *Leserman et al., 2000*). Therefore, transformations of the neurobiological mechanisms in pair bonding are a powerful model for understanding social relationships in our own species, which carries implications for social attachments and romantic partnerships (*Ford and Young, 2021b*; *Volkmar, 2001*; *Young et al., 2008*).

Most of the available knowledge regarding the neurobiology of pair bonding originates from studies on monogamous prairie voles (*Microtus ochrogaster*; *Borie et al., 2022*; *Gobrogge and Wang, 2015*; *Young et al., 2011*). In prairie voles, mesolimbic dopamine (DA), which originates in the ventral tegmental area (VTA), plays an important role in pair bonding (*Curtis and Wang, 2005*). The nucleus accumbens (NAc) shows increased release of DA during sexual behavior (*Aragona et al., 2003*; *Gingrich et al., 2000*), and this increased DA signaling is necessary for the formation of the pair bond. Non-selective DA receptor antagonists in the NAc inhibit mating-induced partner preferences (*Aragona et al., 2003*; *Wang et al., 1999*). Because DA plays an important role in behaviors associated with reward, behavioral reinforcement, and addiction (*Perra et al., 2011*), we predict that the level of real-time DA release in voles may be different depending on whether they encounter their own partner or an unfamiliar vole of the opposite sex.

DA exerts its effect on the formation of the pair bond by binding with its receptors, dopamine type 1 receptor (D1R) and dopamine type 2 receptor (D2R). Moreover, DA affects pair bonding through binding with its receptor in the NAc shell, but not the core. Medium spiny neurons (MSNs) are the main type of efferent neuron in the NAc. NAc MSNs receive dopaminergic inputs to regulate the function of the NAc, which plays an important role in motivation, reward, and drug addiction (*Cooper*

*et al., 2017*; *Floresco, 2015*; *Russo and Nestler, 2013*; *Salgado and Kaplitt, 2015*). MSNs are the main neurons in the NAc to receive, integrate, and transmit information and can be divided into D1 MSNs and D2 MSNs according to different axon projections and distribution of different types of DA receptors (*Gerfen and Surmeier, 2011*; *Surmeier et al., 2007*; *Yager et al., 2015*). Pharmacological studies have shown that D1R and D2R play specific roles in the formation and maintenance of NAc-mediated pair bonding. Blocking D2R with its specific antagonist, while not blocking D1R, prevents pair bonding in prairie voles (*Gingrich et al., 2000*), suggesting the importance of D2R in the formation of the pair bond. In contrast, D1R is involved in the maintenance of the pair bond. D1R is upregulated in the NAc following pair bonding, and D1R signaling helps to maintain partner preference by inducing aggressive behavior toward strangers in paired male prairie voles. The excitability of MSNs is directly related to this change in social behavior. Willett et al. found that in prairie voles, MSNs exhibited notable differences in excitability compared with MSNs in rats; the amplitude of miniature excitatory postsynaptic current (mEPSC) and neuronal excitability decreased with increasing partner preference (*Willett et al., 2018*). A recent study found that cohabitation and mating in female prairie voles can alter both spontaneous and evoked synaptic activity of NAc MSNs; however, which type of MSNs is altered remains unknown (*Borie et al., 2022*). Although D1R and D2R have been found to play different roles in the formation and maintenance of pair bonding using pharmacological methods, whether D1/D2 MSNs display different levels of real-time activity and show different electrophysiological properties after pair bond formation remains unclear.

The ventral pallidum (VP) receives projections from NAc shell D2 MSNs and D1 MSNs (*Kupchik et al., 2015*) and performs a major role in reward, motivation (*Smith et al., 2009*), and partner preference formation (*Lim et al., 2004*). For example, microinjection of a psychostimulant into the VP is sufficient to form a conditioned place preference (*Gong et al., 1996*). VP is also involved in both sexual behavior and social affiliation. In human males, VP activity increases during sexual arousal (*Rauch et al., 1999*). In addition, paired male Callicebus Titi monkeys (*Callicebus nigrifrons*) housed together with their partners show elevated VP glucose metabolism, suggesting that VP neural activity is related to pair bond maintenance (*Bales et al., 2007*). Additionally, inhibition of synaptic transmission from D2 MSNs to the VP can enhance motivation (*Gallo et al., 2018*). Activation of D1 MSNs to the VP mediates aversive, but not reward responses (*Liu et al., 2022*). However, the role of NAc shell D1 and D2 MSNs projecting to the VP in the formation of a pair bond still remains unclear. Therefore, it is important to examine whether NAc shell D2 and D1 MSNs projections to VP play different roles in the formation of the pair bond.

Mandarin voles (*Microtus mandarinus*) are socially monogamous rodents that are widely distributed across China *Li et al., 2021*; they are an ideal animal model for examining the neurobiology of pair bonding. Previous results confirmed that male mandarin voles show significant preference to co-housed partners after 5–7 days of cohabitation (*Li et al., 2021*). Therefore, this study used male mandarin voles and examined the real-time release of DA as well as the activity of two types of MSNs in the NAc shell upon sniffing both their partner and a stranger after pair bonding using in vivo fiber photometry. Alterations in the neural excitability, synaptic plasticity, and sensitivity to DA of NAc shell D1/D2 MSNs after pair bonding were assessed using whole cell patch-clamp recordings and pharmacological methods. Finally, a chemogenetic approach was used to activate or inhibit D2 MSNs and D1 MSNs projections to VP during pair bonding to disclose different roles of these two types of projections in pair bonding. This study advances the understanding of the neurobiological mechanisms underlying the formation of pair bonds and provides insights into the brain mechanism underlying enduring attachment between mating partners.

## Results

### DA release in the NAc shell was higher upon sniffing their partner than sniffing a stranger after pair bonding

After 7 days of cohabitation with females, male mandarin voles showed significant preference for their partners in a partner preference test, displaying reliable pair bonding (*Figure 1—figure supplement 1*). To explore alterations in neural activities induced by the formation of this pair bond, neural activities were measured before (at 3 days of cohabitation) and after (at 7 days of cohabitation) the formation of the pair bond. To determine the changes in DA release after pair bonding, extracellular

DA was monitored by unilateral injection of rAAV-dLight 1.1, which encoded a fluorescent DA sensor into the NAc shell (*Figure 1A–C*). The sites of virus expression and optical fiber layout are shown in *Figure 1B*. The Control group was injected with AAV coding enhanced green fluorescent protein (EGFP). Histological method confirmed that both DA sensor and EGFP were expressed in the NAc shell (*Figure 1B*, *Figure 1—figure supplement 2B*).

Next, the dynamics of endogenous DA were measured in male mandarin voles when they sniffed their partner or a female stranger. Fluorescence signals were found to have increased in the NAc shell relative to baseline after cohabitation for 3 days when male voles sniffed either their partner or an unknown female (*Figure 1—figure supplement 5A*; Paired t test: partner, t (6)=3.290, p=0.0166; stranger, t (6)=4.884, p=0.0028; object, t (6)=1.745, p=0.1315). Signal intensities showed no difference between sniffing their partner or an unknown female after three days cohabitation (*Figure 1D–F*; One-Way Repeated Measures ANOVA: F (2.000, 12.00)=8.702, p=0.0001; partner vs. object, p=0.0060; stranger vs. object, p=0.0239). In addition, the fluorescence signals of the DA release in the NAc shell did not show differences between sniffing partners or strangers after 1 hr or 3 days of cohabitation (*Figure 1—figure supplement 3*; *Figure 1—figure supplement 3F*, One-Way Repeated Measures ANOVA: F (2.000, 12.00)=19.59, p=0.0002; partner vs. object, p=0.0004, stranger vs. object, p=0.0006. *Figure 1—figure supplement 3I*, One-Way Repeated Measures ANOVA: F (2.000, 12.00)=10.85, p=0.0087; partner vs. object, p=0.0056, stranger vs. object, p=0.0044. *Figure 1—figure supplement 3J*, Two-Way Repeated Measures ANOVA: group ×treatment: F (2, 12)=0.5558, p=0.5877; group: F (2, 12)=20.53, p=0.0001; treatment: F (1, 6)=0.1467, p=0.7149). However, after cohabitation for 7 days, although the extracellular DA concentration increased relative to baseline upon sniffing their partner or an unknown female (*Figure 1—figure supplement 5B*; Paired t test: partner, t (6)=6.020, p=0.0009; stranger, t (6)=2.611, p=0.0401; object, t (6)=2.209, p=0.0692), the extracellular DA concentration was significantly higher upon sniffing their partner than a stranger (*Figure 1G–I*; One-Way Repeated Measures ANOVA: F (2.000, 12.00) = 21.41, p=0.0002; partner vs. stranger, p=0.0097; partner vs. object, p<0.0001, stranger vs. object, p=0.0422). DA release did not change upon sniffing an object. In addition, changes in DA release upon sniffing same-sex cagemates and unknown males after cohabitation for 3 and 7 days were also measured. DA release in the NAc shell increased relative to baseline after cohabitation for 3 and 7 days when male voles sniffed an unknown male vole (*Figure 1—figure supplement 6A and B*; A, Paired t test: partner, t (5)=1.464, p=0.2030; stranger, t (5)=3.327, p=0.0209; object, t (5)=1.249, p=0.2669. B, Paired t test: partner, t (5)=1.998, p=0.1021; stranger, t (5)=2.602, p=0.0481; object, t (5)=0.1254, p=0.9051), but no difference was found in signal intensities between sniffing same-sex cagemates and unknown males (*Figure 1—figure supplement 4D–I*; One-Way Repeated Measures ANOVA: F (2.000, 10.00)=4.864, p=0.0335; stranger vs. object, p=0.0396). Carrots are voles' daily food. We found that when voles eat carrots, DA release increases in the NAc shell. The fluorescence signal of DA concentration was examined when voles ate carrot after cohabitation for 3 and 7 days, to rule out that the difference in fluorescence signals was caused by the difference in virus expression at different time points. The results showed that the fluorescence signal in the NAc shell increased upon eating carrot (*Figure 1—figure supplement 7B–D*; Two-Way Repeated Measures ANOVA: group ×treatment: F (1, 7)=0.0274, p=0.8732; group: F (1, 7)=0.0557, p=0.8201; treatment: F (1, 7)=32.79, p=0.0007), but no significant difference in fluorescence signal was found while eating carrot between voles that had cohabitated for 3 and 7 days. In control animals expressing EGFP in the NAc shell, changes in fluorescence intensities showed no difference between male voles sniffing their partner and an unknown female (*Figure 1—figure supplement 2*; E, One-Way Repeated Measures ANOVA: F (2.000, 8.00)=4.092, p=0.0597. G, One-Way Repeated Measures ANOVA: F (2.000, 8.00)=0.6107, p=0.5665. I, One-Way Repeated Measures ANOVA: F (2.000, 8.00)=0.0800, p=0.9238. K, One-Way Repeated Measures ANOVA: F (2.000, 8.00)=2.786, p=0.1207). Furthermore, no significant differences were found between the fluorescence signals of DA release upon side-by-side contact with their partner and a stranger after 3 days of cohabitation (*Figure 1—figure supplement 8A*; Two-tailed unpaired t-test: t (6)=0.08005, p=0.9388). However, the fluorescence signals of DA release upon side-by-side contact with their partner were higher than that during the same behavior with a stranger after 7 days of cohabitation (*Figure 1—figure supplement 8B*; Two-tailed unpaired t-test: t (9)=2.733, p=0.0231).

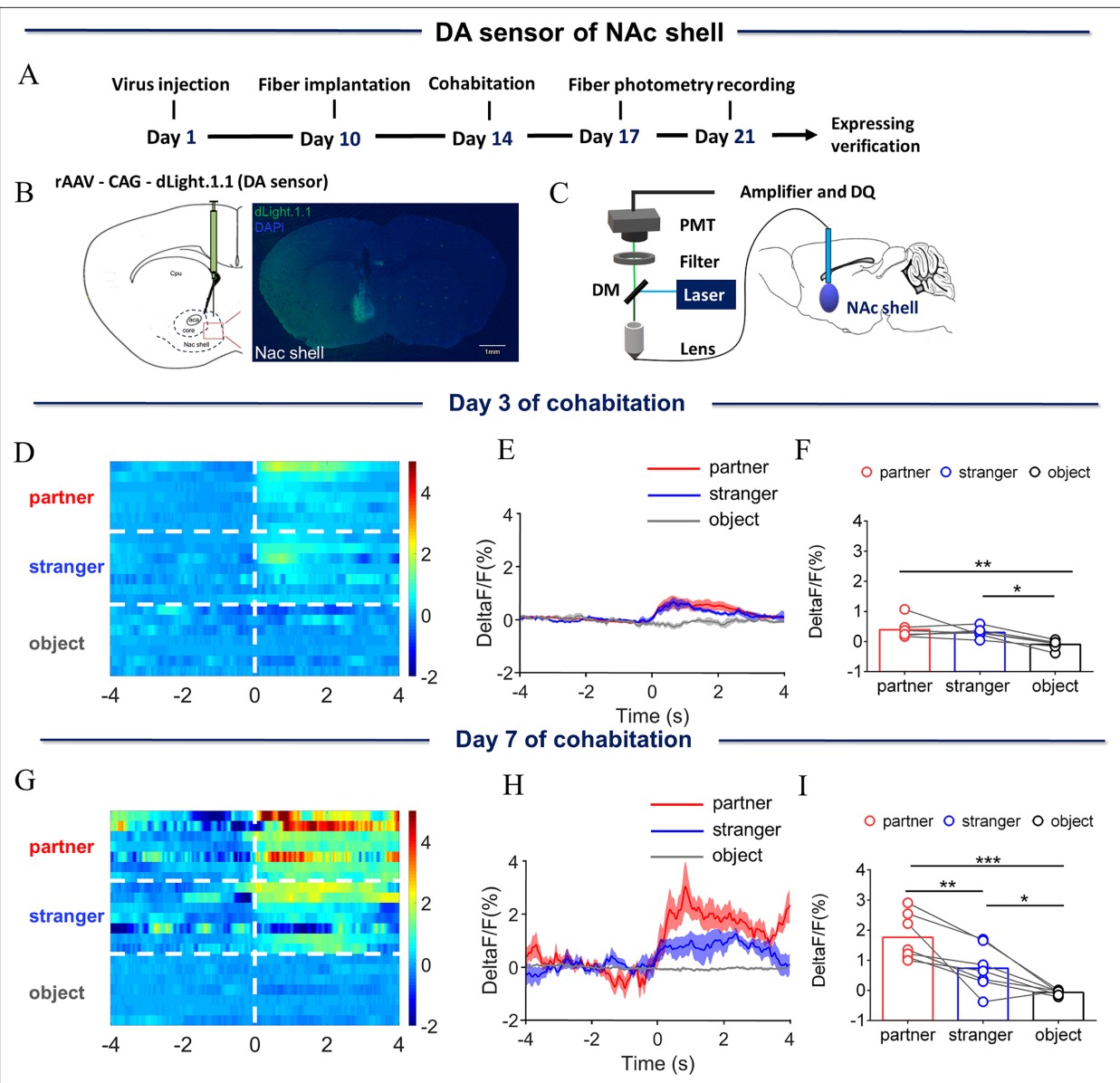

**Figure 1.** Dynamics of extracellular dopamine (DA) concentration within the nucleus accumbens (NAc) shell upon sniffing their partner or an unknown female. (**A**) Timeline of experiments. (**B**) Schematic diagrams depicting virus injection and recording sites and histology showing the expression of DA sensor within the NAc shell. Scale bar: 1 mm. (**C**) Schematic of the procedure used to record extracellular DA concentration in the NAc shell using fiber photometry. (**D**) Heat map illustrating the extracellular DA concentrations (ΔF/F, %) of the NAc shell when sniffing their partner, an unknown female, and an unrelated object. (**E**) Mean fluorescence signal changes of DA sensor during sniffing their partner (red line), an unknown female (blue line), or an object (gray line) after 3 days of cohabitation. The shaded areas along the differently colored lines represent the margin of error. (**F**) Quantification (One-Way Repeated Measures ANOVA) of changes in DA signals during sniffing of their partner, an unknown female, and an object after 3 days of cohabitation. (**G**) Heat map illustrating the extracellular DA concentration (ΔF/F, %) of the NAc shell when sniffing their partner, an unknown female, and an object. (**H**) Mean fluorescence signal changes of the DA sensor when sniffing their partner (red line), an unknown female (blue line), or an object (gray line) after 7 days of cohabitation. (**I**) Quantification (One-Way Repeated Measures ANOVA) of changes in extracellular DA concentration when sniffing their partner, an unknown female, and an object after 7 days of cohabitation. Error bars = SEM * represents p<0.05, ** represents p<0.01, and *** represents p<0.001. See *Figure 1—source data 1* for detailed statistics.

The online version of this article includes the following source data and figure supplement(s) for figure 1:

**Source data 1.** Raw data and statistical results of changes in DA signals during sniffing their partner, an unknown female, and an object after cohabitation.

**Figure supplement 1.** Mandarin voles showed significant preference to partners in the partner preference test after cohabitation.

**Figure supplement 1—source data 1.** Raw data and statistical results of side-by-side time in the partner preference test after cohabitation for 3 days

*Figure 1 continued on next page*

*Figure 1 continued*

and 7 days.

**Figure supplement 2.** NAc shell EGFP fluorescence signals (with injection of control virus) upon sniffing partner or an unknown female after cohabitation.

**Figure supplement 2—source data 1.** Raw data and statistical results of changes in EGFP fluorescence signals during sniffing after cohabitation.

**Figure supplement 3.** Dynamics of extracellular dopamine (DA) concentration within the nucleus accumbens (NAc) shell upon sniffing their partner or an unknown female after 1 hr or 3 days of cohabitation.

**Figure supplement 3—source data 1.** Raw data and statistical results of changes in DA signals during sniffing of their partner, an unknown female, and an object after cohabitation.

**Figure supplement 4.** Dynamics of extracellular DA concentration within the NAc shell upon sniffing their cagemate or an unknown male.

**Figure supplement 4—source data 1.** Raw data and statistical results of changes in DA signals during sniffing of their cagemate, an unknown male, and an object after cohabitation.

**Figure supplement 5.** NAc shell Dynamics of extracellular DA concentration when sniffing their partner or an unknown female after cohabitation.

**Figure supplement 5—source data 1.** Raw data and statistical results of changes in DA signals during sniffing their partner, an unknown female, and an object after cohabitation.

**Figure supplement 6.** NAc shell Dynamics of extracellular DA concentration when sniffing their cagemate or an unknown male after cohabitation.

**Figure supplement 6—source data 1.** Raw data and statistical results of changes in DA signals during sniffing of their cagemate, an unknown male, and an object after cohabitation.

**Figure supplement 7.** NAc shell dynamics of extracellular DA concentration upon eating carrot.

**Figure supplement 7—source data 1.** Raw data and statistical results of changes in DA signals upon eating carrot.

**Figure supplement 8.** NAc shell Dynamics of extracellular DA release during side-by-side contact after cohabitation.

**Figure supplement 8—source data 1.** Raw data and statistical results of changes in DA signals during side-by-side contact after cohabitation.

**Figure supplement 9.** Quantification of fluorescence change during non-social behavioral bout and sniffing voles after cohabitation.

**Figure supplement 9—source data 1.** Raw data and statistical results of changes in DA signals during non-social behavioral bouts and sniffing voles after cohabitation.

## D2 MSNs activity in the NAc shell decreased while D1 MSNs activity increased upon sniffing their partner after pair bonding

The above results provide evidence that DA released within the NAc shell may play an important role in the formation of the pair bond among mandarin voles. Whether MSNs display different activities in vivo after the formation of partner preference was examined next. Fourteen days prior to cohabitation, rAAV-D1/D2-GCaMP6m, a D1/D2 genetically encoded fluorescent calcium sensor, was injected into the NAc shell. Dynamics of real-time calcium signals in the NAc shell in freely moving voles were recorded via a fiber photometry system in pair-bonded or non-bonded mandarin voles (*Figures 2A, B, 3A and B*). Immunofluorescence images showed that 84.42% of D1-GCaMP6m cells and 85.96% of D2-GCaMP6m cells were D1R-mRNA and D2R-mRNA positive (*Figures 2C and 3C*).

The results showed that the fluorescence signal of D1/D2 MSNs in the NAc shell was significantly increased relative to baseline when eating carrot after cohabitation for 3 and 7 days; voles who had experienced cohabitation for 3 and 7 days displayed no significant difference in fluorescence signals upon eating carrot (*Figure 2—figure supplement 5*, *Figure 3—figure supplement 5*) (*Figure 2—figure supplement 5*: Two-Way Repeated Measures ANOVA: group ×treatment: $F_{(1, 5)}=0.7013$, p=0.4405; group: $F_{(1, 5)}=0.6822$, p=0.4464; treatment: $F_{(1, 5)}=42.67$, p=0.0013. *Figure 3—figure supplement 5*: Two-Way Repeated Measures ANOVA: group ×treatment: $F_{(1, 7)}=1.1413$, p=0.2733; group: $F_{(1, 7)}=1.428$, p=0.2711; treatment: $F_{(1, 7)}=8.499$, p=0.0225.). D2 MSNs activity was significantly reduced relative to baseline when sniffing either their partner or an unknown female after 3 and 7 days of cohabitation (*Figure 2—figure supplement 3A and B*) (C, Paired t test: partner, $t_{(6)}=2.936$, p=0.0261; stranger, $t_{(6)}=2.499$, p=0.0466; object, $t_{(6)}=0.8186$, p=0.4443. D, Paired t test: partner, $t_{(6)}=3.792$, p=0.0091; stranger, $t_{(6)}=3.935$, p=0.0077; object, $t_{(6)}=0.1841$, p=0.8600). In contrast, after 7 days of cohabitation, the activity when sniffing their partner was significantly lower than when sniffing a stranger (*Figure 2G–I*; One-Way Repeated Measures ANOVA: $F_{(1.212, 7.274)}=15.89$, p=0.0039; partner vs. stranger, p=0.0366; partner vs. object, p=0.0140, stranger vs. object, p=0.0434). After 3 days of cohabitation, D2 MSNs activity significantly decreased relative to

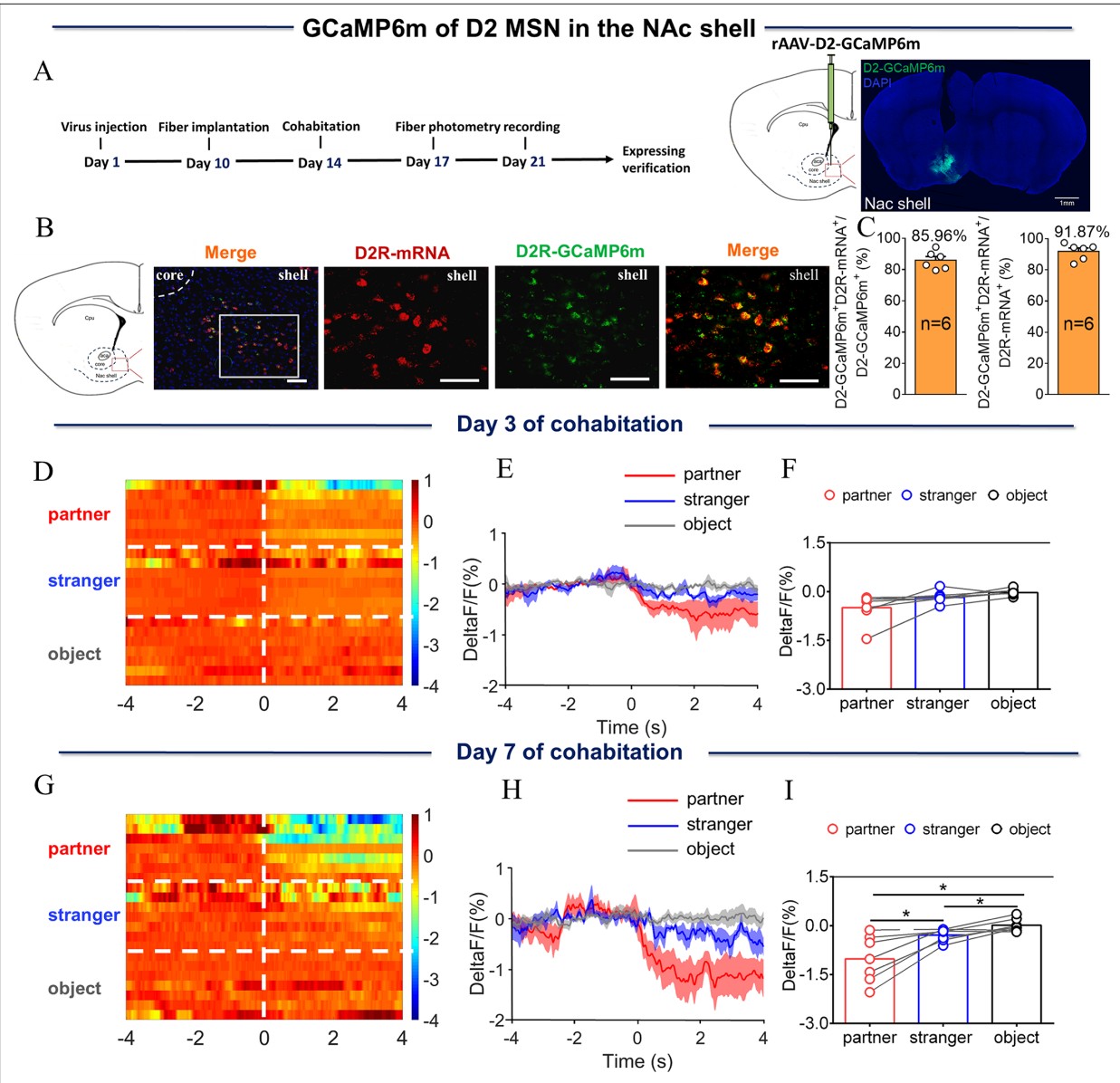

**Figure 2.** NAc shell D2 medium spinous neurons (MSNs) showing decreased activity upon sniffing their partner or an unknown female after cohabitation. (**A**) Left: Timeline of experiments; right: Schematic diagrams depicting virus injection and recording sites as well as histology showing the expression of D2-GCaMP6m within the NAc shell. Scale bar: 1 mm. (**B**) Overlaps of D2-GCaMP6m (green), D2R-mRNA (red), and DAPI (blue) in the NAc shell. Scale bar: 100 μm. (**C**) Statistical chart showing that D2-GCaMP6m was relatively restricted to D2R-mRNA positive neurons (n=6 voles). (**D**) Heat map illustrating the calcium response (ΔF/F, %) of the NAc shell when sniffing a partner, an unknown female, or an object after cohabitation for 3 days. (**E, H**) Mean fluorescence signal changes of the calcium response during sniffing their partner (red line), an unknown female (blue line), or an object (gray line) after cohabitation for 3 days (**E**) and 7 days (**H**). The shaded areas along the different colors of lines show the margins of error. (**F, I**) Quantification (One-Way Repeated Measures ANOVA) of changes in calcium signals when sniffing their partner, an unknown female, or an object after cohabitation for 3 days (**F**) (n=7 voles) and 7 days (**I**) (n=7 voles). (**G**) Heat map illustrating calcium signals (ΔF/F, %) of the NAc shell when sniffing their partner, an unknown female, or an object after cohabitation for 7 days. All error bars = SEM. * represents p<0.05. See *Figure 2—source data 1* for detailed statistics.

The online version of this article includes the following source data and figure supplement(s) for figure 2:

**Source data 1.** Raw data and statistical results of changes in calcium signals during sniffing of their partner, an unknown female, and an object after cohabitation.

**Figure supplement 1.** NAc shell D2 EGFP MSNs fluorescence signals after cohabitation.

**Figure supplement 1—source data 1.** Raw data and statistical results of changes in D2-EGFP fluorescence signals during sniffing after cohabitation.

*Figure 2 continued on next page*

*Figure 2 continued*

**Figure supplement 2.** NAc shell D2 MSNs showing decreased activity upon sniffing their cagemate or an unknown male after cohabitation.

**Figure supplement 2—source data 1.** Raw data and statistical results of changes in fluorescence signals during sniffing of their cagemate, an unknown male, and an object after cohabitation.

**Figure supplement 3.** NAc shell Dynamics of D2 MSNs when sniffing their partner or an unknown female after cohabitation.

**Figure supplement 3—source data 1.** Raw data and statistical results of changes in calcium signals during sniffing their partner, an unknown female, and an object after cohabitation.

**Figure supplement 4.** NAc shell Dynamics of D2 MSNs when sniffing their cagemate or an unknown male after cohabitation.

**Figure supplement 4—source data 1.** Raw data and statistical results of changes in calcium signals during sniffing their cagemate, an unknown male, and an object after cohabitation.

**Figure supplement 5.** NAc shell Dynamics of D2 MSNs upon eating carrot.

**Figure supplement 5—source data 1.** Raw data and statistical results of changes in calcium signals upon eating carrot.

**Figure supplement 6.** NAc shell Dynamics of D2 MSNs during side-by-side contact after cohabitation.

**Figure supplement 6—source data 1.** Raw data and statistical results of changes in calcium signals during side-by-side contact after cohabitation.

**Figure supplement 7.** Quantification of fluorescence change during non-social behavioral bout and sniffing voles after cohabitation.

**Figure supplement 7—source data 1.** Raw data and statistical results of changes in calcium signals during non-social behavioral bouts and sniffing voles after cohabitation.

**Figure supplement 8.** The supplement FISH image of the D2-GCaMP6m MSNs in the NAc shell.

**Figure supplement 8—source data 1.** Counts of D2-GCaMP6m (green) positive neurons, D1R-mRNA (red) positive neurons, and co-expressed neurons.

baseline when sniffing an unknown male (*Figure 2—figure supplement 4A*; Paired t test: partner, t (5)=2.479, p=0.00559; stranger, t (5)=2.654, p=0.0452; object, t (5)=0.5164, p=0.6276). After 7 days of cohabitation, D2 MSNs activity significantly decreased relative to baseline when sniffing a same-sex cagemate or an unknown male (*Figure 2—figure supplement 4B*; Paired t test: partner, t (5)=3.267, p=0.0223; stranger, t (5)=2.789, p=0.0385; object, t (5)=1.035, p=0.3479). However, no significant difference in fluorescence signals was found when sniffing a same-sex cagemate or an unknown male (*Figure 2—figure supplement 2G–I*; One-Way Repeated Measures ANOVA: F (2.000, 10.00)=4.940, p=0.0322; stranger vs. object, p=0.0426). However, D1 MSNs activity was significantly increased relative to baseline when sniffing their partner or an unknown female after 3 days of cohabitation (*Figure 3—figure supplement 3A and B*; A, Paired t test: partner, t (6)=2.700, p=0.0356; stranger, t (6)=2.492, p=0.0470; object, t (6)=0.5216, p=0.6207. B, Paired t test: partner, t (6)=3.591, p=0.0115; stranger, t (6)=2.381, p=0.0547; object, t (6)=1.590, p=0.1630). After 7 days of cohabitation, D1 MSNs activity significantly increased relative to baseline when sniffing their partner and was significantly higher when sniffing their partner than when sniffing an unknown female (*Figure 3G–I*; One-Way Repeated Measures ANOVA: F (1.242, 7.451)=11.11, p=0.0093; partner vs. stranger, p=0.0278, partner vs. object, p=0.0326). After cohabitation for 3 or 7 days, D1 MSNs activity significantly increased relative to baseline when sniffing an unknown male (*Figure 3—figure supplement 4A and B*; A, Paired t test: partner, t (5)=1.788, p=0.1338; stranger, t (5)=2.613, p=0.0475; object, t (5)=0.4349, p=0.6818. B, Paired t test: partner, t (5)=2.009, p=0.1008; stranger, t (5)=2.983, p=0.0307; object, t (5)=1.049, p=0.3420). No significant difference in fluorescence signals was found upon sniffing a same-sex cagemate or an unknown male (*Figure 3—figure supplement 2G–I*; One-Way Repeated Measures ANOVA: F (2.000, 10.00)=5.533, p=0.0241; stranger vs. object, p=0.0278). D2 and D1 MSNs activity did not change upon sniffing an object. In addition, no significant changes were detected in the fluorescence signal upon sniffing their partner or a stranger in D1/D2 MSNs of voles injected with control virus (EGFP) without GCaMP6m sequence in the construct (*Figure 2—figure supplements 1*) (*Figure 2—figure supplement 1E*, One-Way Repeated Measures ANOVA: F (2.000, 8.00)=0.7086, p=0.5208. *Figure 2—figure supplement 1G*, One-Way Repeated Measures ANOVA: F (2.000, 8.00)=1.450, p=0.2901. *Figure 2—figure supplement 1I*, One-Way Repeated Measures ANOVA: F (2.000, 8.00)=0.7471, p=0.5041. *Figure 2—figure supplement 1K*, One-Way Repeated Measures ANOVA: F (2.000, 8.00)=1.344, p=0.3140. *Figure 3—figure supplement 1E*, One-Way Repeated Measures ANOVA: F (2.000, 8.00)=4.162, p=0.0577. *Figure 3—figure supplement 1G*, One-Way Repeated Measures ANOVA: F (2.000, 8.00)=0.6909, p=0.5287. *Figure 3—figure*

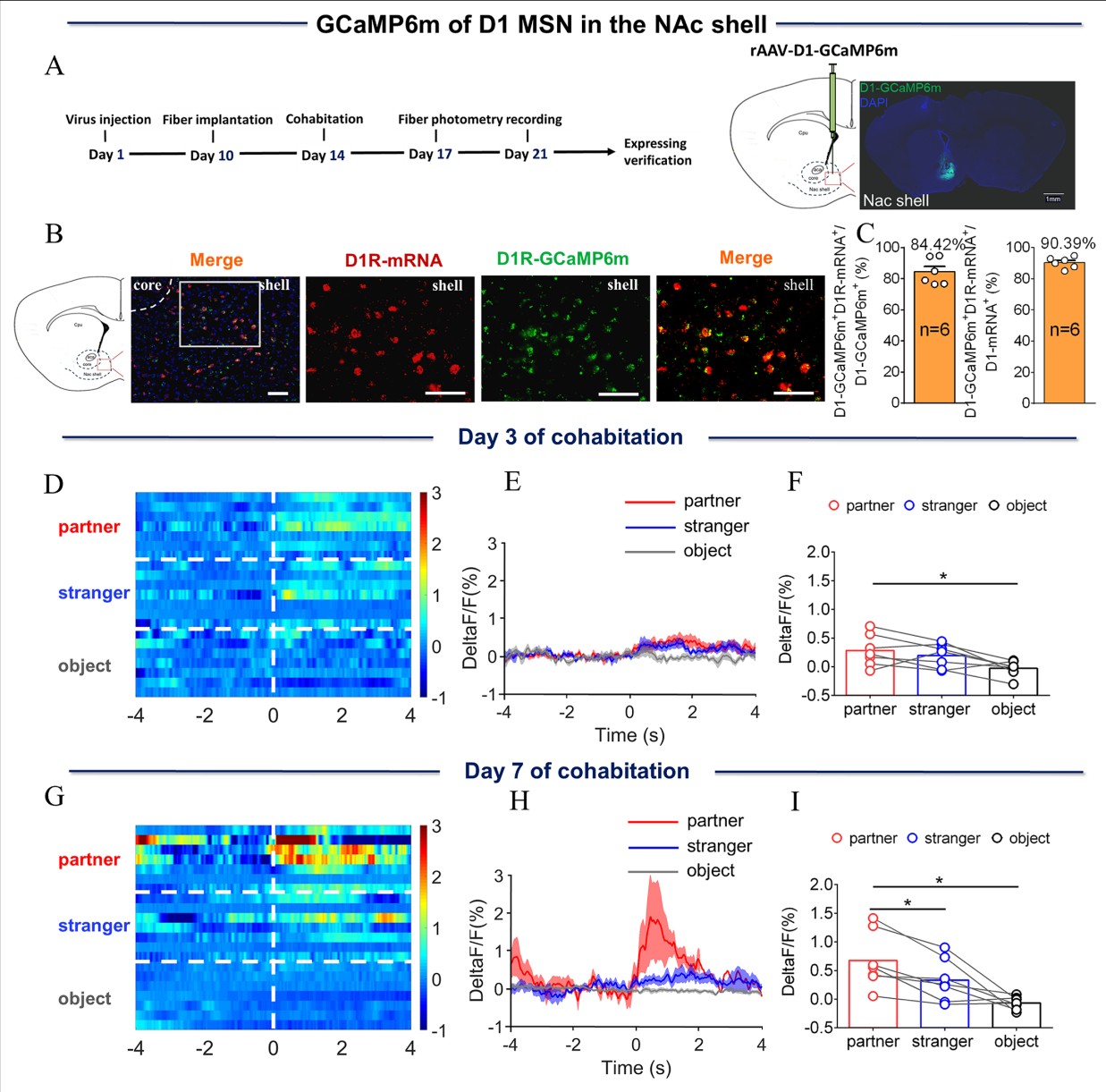

**Figure 3.** NAc shell D1 MSNs showing increased activity when sniffing their partner or an unknown female after cohabitation. (**A**) Left: Timeline of experiments; right: Schematic diagrams depicting the virus injection and recording sites as well as histology showing the expression of D1-GCaMP6m within the NAc shell. Scale bar: 1 mm. (**B**) Overlap of D1-GCaMP6m (green), D1R-mRNA (red), and DAPI (blue) in the NAc shell. Scale bar: 100 μm. (**C**) Statistical chart showing that D1-GCaMP6m was relatively restricted to D1R-mRNA positive neurons (n=6 voles). (**D**) Heat map illustrating the calcium signals (ΔF/F, %) of the NAc shell when sniffing their partner, an unknown female, or an object after cohabitation for 3 days. (**E, H**) Mean fluorescence changes of calcium signals when sniffing their partner (red line), an unknown female (blue line), or an object (gray line) after cohabitation for 3 days (**E**) and 7 days (**H**). The shaded area along the different colored lines represents error margins. (**F, I**) Quantification (One-Way Repeated Measures ANOVA) of changes in calcium signals when sniffing their partner, an unknown female, or an object after cohabitation for 3 days (**F**) (n=7 voles) and 7 days (**I**) (n=7 voles). (**G**) Heat map illustrating the calcium signals (ΔF/F, %) of NAc shell when sniffing their partner, an unknown female, or an object after cohabitation for 7 days. All error bars = SEM. * represents p<0.05. See *Figure 3—source data 1* for detailed statistics.

The online version of this article includes the following source data and figure supplement(s) for figure 3:

**Source data 1.** Raw data and statistical results of changes in calcium signals during sniffing their partner, an unknown female, and an object after cohabitation.

**Figure supplement 1.** NAc shell D1 EGFP MSNs fluorescence signal with injection of control virus after cohabitation.

**Figure supplement 1—source data 1.** Raw data and statistical results of changes in D1-EGFP fluorescence signals during sniffing after cohabitation.

**Figure supplement 2.** NAc shell D1 MSNs showing increased activity when sniffing their cagemate or an unknown male after cohabitation.

*Figure 3 continued on next page*

*Figure 3 continued*

**Figure supplement 2—source data 1.** Raw data and statistical results of changes in fluorescence signals during sniffing their cagemate, an unknown male, and an object after cohabitation.

**Figure supplement 3.** NAc shell Dynamics of D1 MSNs when sniffing their partner or an unknown female after cohabitation.

**Figure supplement 3—source data 1.** Raw data and statistical results of changes in calcium signals during sniffing of their partner, an unknown female, and an object after cohabitation.

**Figure supplement 4.** NAc shell Dynamics of D1 MSNs when sniffing their cagemate or an unknown male after cohabitation.

**Figure supplement 4—source data 1.** Raw data and statistical results of changes in calcium signals during sniffing of their cagemate, an unknown male, and an object after cohabitation.

**Figure supplement 5.** NAc shell Dynamics of D1 MSNs upon eating carrot.

**Figure supplement 5—source data 1.** Raw data and statistical results of changes in calcium signals upon eating carrot.

**Figure supplement 6.** NAc shell Dynamics of D1 MSNs during side-by-side contact after cohabitation.

**Figure supplement 6—source data 1.** Raw data and statistical results of changes in calcium signals during side-by-side contact after cohabitation.

**Figure supplement 7.** Quantification of fluorescence change during non-social behavioral bout and sniffing voles after cohabitation.

**Figure supplement 7—source data 1.** Raw data and statistical results of changes in calcium signals during non-social behavioral bouts and sniffing voles after cohabitation.

**Figure supplement 8.** The supplement FISH image of the D1-GCaMP6m MSNs in the NAc shell.

**Figure supplement 8—source data 1.** Counts of D1-GCaMP6m (green) positive neurons, D2R-mRNA (red) positive neurons, and co-expressed cells.

*supplement 1I*, One-Way Repeated Measures ANOVA: F (2.000, 8.00)=0.5571, p=0.5936. *Figure 3—figure supplement 1K*, One-Way Repeated Measures ANOVA: F (2.000, 8.00)=2.178, p=0.1757). Similarly, the fluorescence signals of D2 MSNs and D1 MSNs upon side-by-side contact with their partner or an unknown female after 3 days of cohabitation did not differ. However, the fluorescence signals of D2 MSNs and D1 MSNs upon side-by-side contact with their partner were higher than that upon the same behavior toward an unknown female after 7 days of cohabitation (*Figure 2—figure supplement 6*, *Figure 3—figure supplement 6*; *Figure 2—figure supplement 6A*: Two-tailed paired t-test, t (6)=0.6936, p=0.5139; *Figure 2—figure supplement 6B*: Two-tailed unpaired t-test, t (10)=2.813, p=0.0184; *Figure 3—figure supplement 6A*: Two-tailed unpaired t-test, t (6)=1.943, p=0.1000; *Figure 3—figure supplement 6B*: Two-tailed unpaired t-test, t (7)=2.392, p=0.0481).

## Cohabitation with a partner alters the electrophysiological properties and synaptic transmission of D2/D1 MSNs in the NAc shell

DA binding with its target receptor (D2R) in the NAc is necessary for the formation of a pair bond. D2 MSNs were identified by infection of the NAc shell with rAAV-D2-mCherry virus, and changes in synaptic transmission in D2 MSNs were recorded using whole-cell patch-clamp (*Figure 4A*). The frequency and amplitude of D2 MSNs spontaneous excitatory postsynaptic current (sEPSC) were found to be significantly higher in paired males than in naive males (*Figure 4B and C*; Two-tailed unpaired t-test: frequency: t (20)=–3.634, p=0.002; amplitude: t (20)=–2.619, p=0.016). In addition, the frequency, but not the amplitude, of D2 MSNs spontaneous inhibitory postsynaptic currents (sIPSCs) was significantly increased in paired males compared to naive males (*Figure 4B and D*) (Two-tailed unpaired t-test: frequency: t (26)=–2.096, p=0.046; amplitude: t (26)=–0.528, p=0.602). Synapse-driven homeostatic plasticity of intrinsic excitability—long-term changes in synaptic activity that modify intrinsic excitability—has been found in the NAc shell. However, this study failed to detect a significant change in intrinsic excitability in D2 MSNs in both naive males and paired males (*Figure 4E–H*) (4 F: Two-Way Repeated Measures ANOVA, group ×treatment: F (8, 198)=0.070, p=0.999; group: F (1, 198)=0.0003, p=0.986; treatment: F (8, 198)=16.767, p<0.0001; 4 G, Two-Way Repeated Measures ANOVA, group ×treatment: F (8, 198)=0.103, p=0.999; group: F (1, 198)=0.078, p=0.780; treatment: F (8, 198)=10.095, p<0.0001; 4 H, Two-tailed unpaired t-test, t (22)=0.100, p=0.921). To further examine the changes in excitatory and inhibitory synaptic transmission of D2 MSNs, evoked excitatory and inhibitory postsynaptic currents (PSCs) were recorded from the same cells. An increased E/I ratio of PSCs was found in D2 MSNs of paired males (*Figure 4J*) (Two-tailed unpaired t-test: t (10)=–3.499, p=0.012). Finally, whether the sEPSC of D2 MSNs could be evoked by bath-applied DA (5 μM) in naive and paired males was also tested. The results showed a significant decrease in both the frequency and

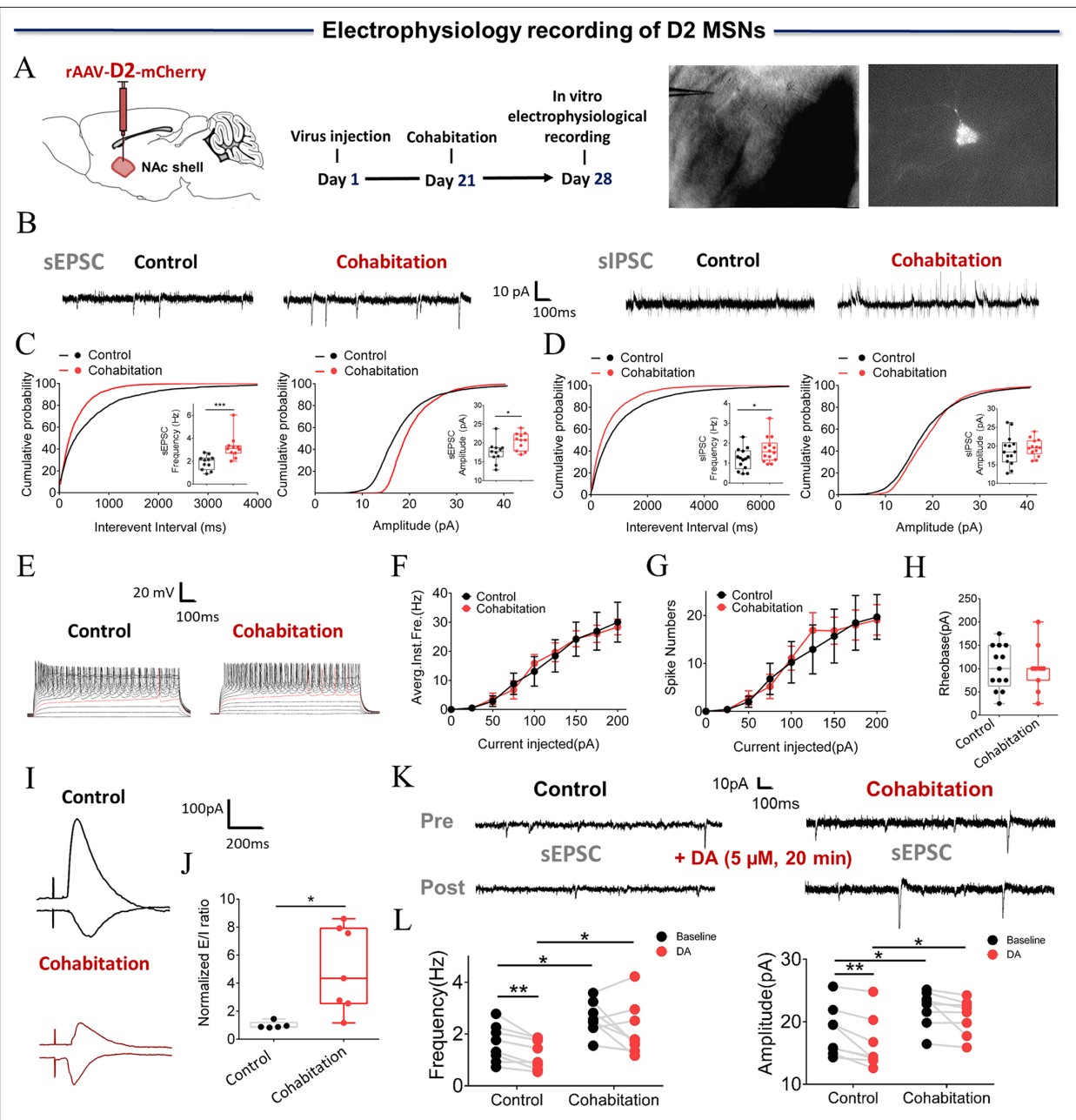

**Figure 4.** Synaptic transmission and neuronal excitability of D2 MSNs in the NAc shell change after cohabitation. (**A**) Timeline of experiments (left), schematic diagrams depicting virus injection and recording sites (middle), and a D2-positive neuron with a micropipette (right). (**B**) Representative spontaneous excitatory postsynaptic current (sEPSC) and spontaneous inhibitory postsynaptic currents (sIPSC) traces from paired and naive voles. (**C**) NAc shell D2 MSNs in paired voles exhibited sEPSCs with higher frequencies (Cohabitation: n=11 cells from four voles; Control: n=11 cells from four voles) and peak amplitudes (Cohabitation: n=11 cells from four voles; Control: n=11 cells from four voles) than those observed in naive voles. (**D**) NAc shell D2 MSNs in paired voles exhibited sIPSCs with higher frequencies (Cohabitation: n=14 cells from four voles; Control: n=14 cells from four voles) than those observed in naive voles. (Amplitude: Cohabitation: n=14 cells from four voles; Control: n=14 cells from four voles) (**E–H**) The neuronal excitability of D2 MSNs in the NAc shell of paired voles was similar to that of naive voles. (Cohabitation: n=11 cells from four voles; Control: n=13 cells from six voles). (**I and J**) The excitation-inhibition ratio was higher in paired voles than in naive voles. (Cohabitation: n=7 cells from four voles; Control: n=5 cells from four voles). (**K and L**) D2-sEPSCs are evoked over a background of bath-applied DA in naive voles and cohabitated (7 days) voles. (Cohabitation: n=8 cells from four voles; Control: n=8 cells from three voles). Error bars = SEM. * represents p<0.05, ** represents p<0.01, and *** represents p<0.001. See *Figure 4—source data 1* for detailed statistics.

The online version of this article includes the following source data for figure 4:

**Source data 1.** Raw data and statistical results of sEPSC, sIPSC, neuronal excitability, and excitation-inhibition ratio in the NAc shell D2 MSNs in naive voles and cohabitated (7 days) voles.

amplitude of D2 sEPSC in naive (*Figure 4K and L*; Two-Way Repeated Measures ANOVA: frequencies: group ×treatment: F (1, 14)=0.0177, p=0.896; group: F (1, 14)=8.105, p=0.0129; treatment: F (1, 14)=7.189, p=0.0179, Control_ Baseline vs Control DA, p=0.0050, Cohabitation _ Baseline vs Cohabitation DA, p=0.2296, Control_ Baseline vs Cohabitation Baseline, p=0.0277, Control _DA vs Cohabitation DA, p=0.0376; amplitude: group ×treatment: F (1, 14)=0.7937, p=0.4042; group: F (1, 14)=5.911, p=0.0291; treatment: F (1, 14)=14.78, p=0.0018, Control_ Baseline vs Control DA, p=0.007, Cohabitation Baseline vs Cohabitation _DA, p=0.098, Control Baseline vs Cohabitation Baseline, p=0.0472, Control DA vs Cohabitation DA, p=0.0241), but not in paired mandarin voles (*Figure 4L*).

Pharmacological assessment showed that D1R is involved in the maintenance of the pair bond; therefore, whole-cell patch-clamp recording was performed to determine whether D1 MSNs synaptic transmission in the NAc shell was altered by cohabitation. D1 MSNs showed reduced frequency and amplitude of sEPSC after cohabitation *Figure 5B and C*; (Two-tailed unpaired t-test: frequency: t (33)=2.816, p=0.008; amplitude: t (33)=3.322, p=0.002). In addition, in sIPSC in D1 MSNs, the frequency decreased drastically, but not the amplitude (*Figure 5B and D*; Two-tailed unpaired t-test: frequency: t (24)=2.324, p=0.024; amplitude: t (24)=0.100, p=0.921). Next, the intrinsic excitability was examined between naive and paired males, and a significant increase in intrinsic excitability of D1 MSNs was found (*Figure 5E–H*; 5 F: Two-Way Repeated Measures ANOVA, group ×treatment: F (8, 126)=0.778, p=0.623; group: F (1, 126)=7.757, p=0.006; treatment: F (8, 126)=29.810, p<0.0001; 5 G, Two-Way Repeated Measures ANOVA, group ×treatment: F (8, 126)=0.845, p=0.565; group: F (1, 126)=5.440, p=0.021; treatment: F (8, 126)=15.757, p<0.0001; 5 H, Two-tailed unpaired t-test, t (14)=2.744, p=0.016); differences in the E/I ratio of D1 MSNs were not found (*Figure 5J*; Two-tailed unpaired t-test: t (12)=−1.429, p=0.179). Then, it was tested whether the sEPSC of D1 MSNs can be evoked by bath-applied DA in naive and paired males. The results showed a significant increase in the frequency of D1 sEPSC in naïve males (*Figure 5K and L*; Two-Way Repeated Measures ANOVA: frequencies: group ×treatment: F (1, 18)=4.784, p=0.0422; group: F (1, 18)=9.954, p=0.0055; treatment: F (1, 18)=5.164, p=0.0356, Control_ Baseline vs Control DA, p=0.006, Cohabitation _ Baseline vs Cohabitation DA, p=0.953, Control Baseline vs Cohabitation _ Baseline, p=0.015, Control DA vs Cohabitation DA, p=0.006; amplitude: group ×treatment: F (1, 18)=0.007, p=0.9343; group: F (1, 18)=5.842, p=0.0265; treatment: F (1, 18)=0.6498, p=0.4307, Control_ Baseline vs Control DA, p=0.5285, Cohabitation Baseline vs Cohabitation DA, p=0.63468, Control Baseline vs Cohabitation Baseline, p=0.0282, Control DA vs Cohabitation DA, p=0.029), but not in paired males (*Figure 5L*).

## Effects of chemogenetic activation or inhibition of D2/D1 MSNs in the NAc shell projecting to the VP upon the formation of partner preference

MSNs in the NAc shell project extensively to the VP. To activate or inhibit these projections using a chemogenetic approach, rAAV-DIO-hM3Dq-mCherry or rAAV-DIO-hM4Di-mCherry were injected into the NAc shell and rAAV (Retro)-D2-Cre was injected into the VP to selectively express 'Gq-DREADD' or 'Gi-DREADD' in NAc shell VP-projecting D2 MSNs (*Figure 6A*). Immunohistochemical staining showed that 90.11% of mCherry cells co-expressed D2R (*Figure 6B, C*, *Figure 6—figure supplement 1*). To determine whether the ligand clozapine N-oxide (CNO) could activate or inhibit VP-projecting D2 MSNs, whole cell patch-clamp recordings were performed. The results showed that the addition of 10 μM CNO remarkably increased the number of action potentials in the Gq-DREADD-transfected neurons (*Figures 6D and 7D*). In contrast, CNO caused a significant decrease in membrane potentials in Gi-DREADD-transfected neurons (*Figures 6D and 7D*). These results signal the specificity and validation of this virus strategy.

During cohabitation, CNO (1 mg/kg) or saline was intraperitoneally injected daily (*Figure 6A*). The partner preference test was conducted on days 3 and 7 of cohabitation. Subsequent behavioral studies showed that CNO-treated mandarin voles with D2 MSNs Gi-DREADD virus infection showed increased side-by-side alignment with their partners compared with strangers in a partner preference test (*Figure 6E and G*; 6E, Two-way ANOVA: group ×treatment: F (2, 40)=4.942, p=0.012; group: F (2, 40)=4.265, p=0.021; treatment: F (1, 40)=4.967, p=0.032; Gi_Saline vs Gi_CNO, p<0.001; Gq_Saline vs Gq_CNO, p=0.380; mCherry_Saline vs mCherry_CNO, p=0.052. 6 G, Kruskal-Wallis: H=15.198, p=0.002, CNO_Partner vs CNO_Stranger, p<0.0001; saline_Partner vs saline_Stranger, p=0.328); this shows that inhibition of VP-projecting D2 MSNs prompted the formation of partner

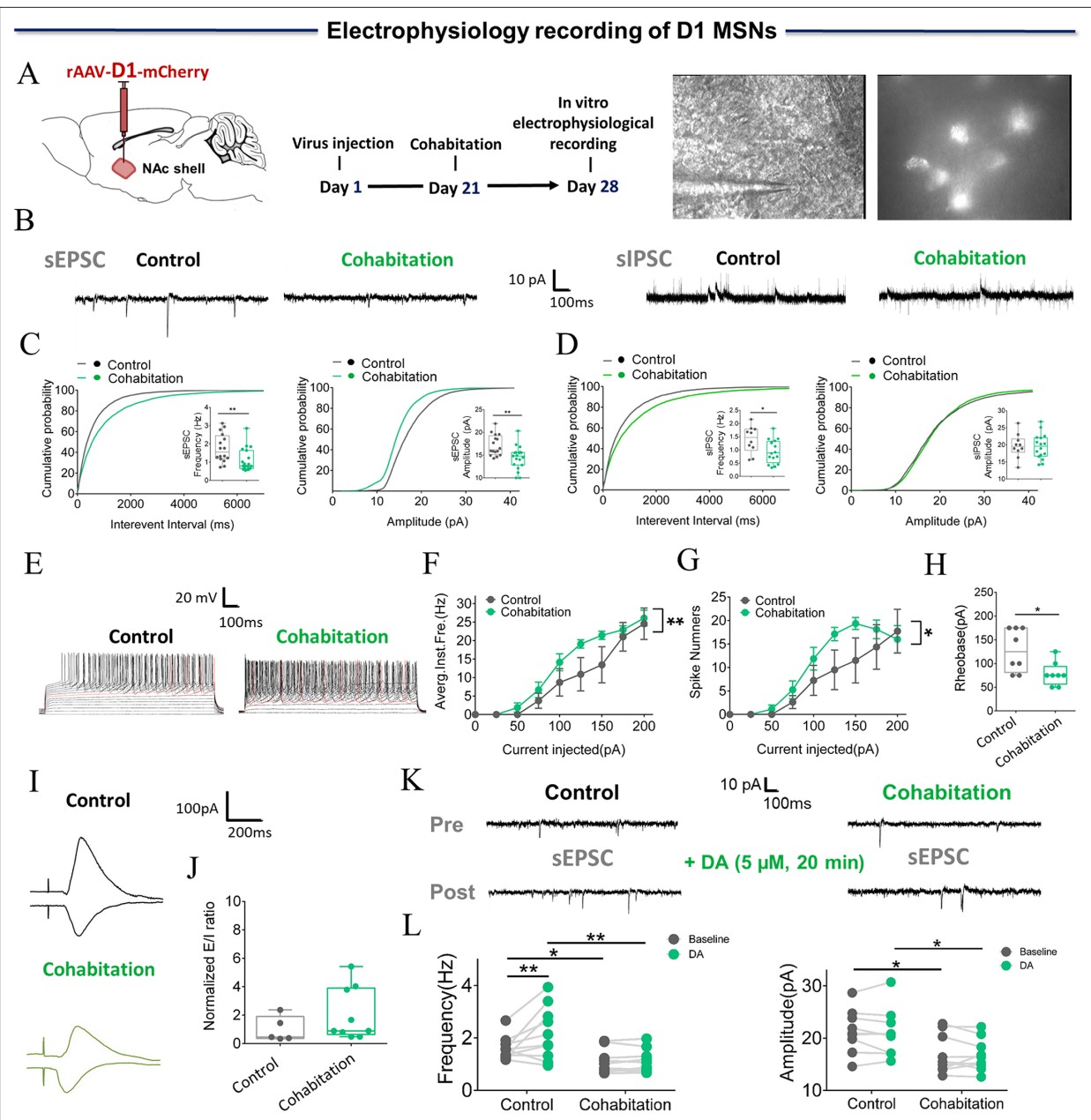

**Figure 5.** Synaptic transmission and neuronal excitability of D1 MSNs in NAc shell change after cohabitation. (**A**) Timeline of experiments (left), schematic diagrams depicting virus injection and recording sites (middle), and a D1-positive neuron with a micropipette (right). (**B**) Representative sEPSC and sIPSC traces from paired and naive voles. (**C**) NAc shell D1 MSNs in paired voles exhibited sEPSCs with lower frequencies (Cohabitation: n=19 cells from five voles; Control: n=16 cells from four voles) and peak amplitudes (Cohabitation: n=19 cells from five voles; Control: n=16 cells from four voles) than those observed in naive voles. (**D**) NAc shell D1 MSNs in paired voles exhibited sIPSCs with lower frequencies (Cohabitation: n=16 cells from four voles; Control: n=10 cells from three voles) than those observed in naive voles (amplitude: Cohabitation: n=16 cells from four voles; Control: n=10 cells from three voles). (**E–H**) Neuronal excitability of D1 MSNs in the NAc shell of paired voles was higher than those observed in naive voles (Cohabitation: n=8 cells from three voles; Control: n=8 cells from four voles). (**I, J**) Excitation-inhibition ratio of paired voles was similar to naive voles. (Cohabitation: n=9 cells from four voles; Control: n=5 cells from three voles). (**K, L**) D1-sEPSCs are evoked over a background of bath-applied DA in naive voles and cohabitated (7 days) voles. (Cohabitation: n=10 cells from five voles; Control: n=10 cells from four voles). Error bars = SEM. * represents p<0.05, and ** represents p<0.01. See *Figure 5—source data 1* for detailed statistics.

The online version of this article includes the following source data for figure 5:

**Source data 1.** Raw data and statistical results of sEPSC, sIPSC, neuronal excitability, and excitation-inhibition ratio in the NAc shell D1 MSNs in naive voles and cohabitated (7 days) voles.

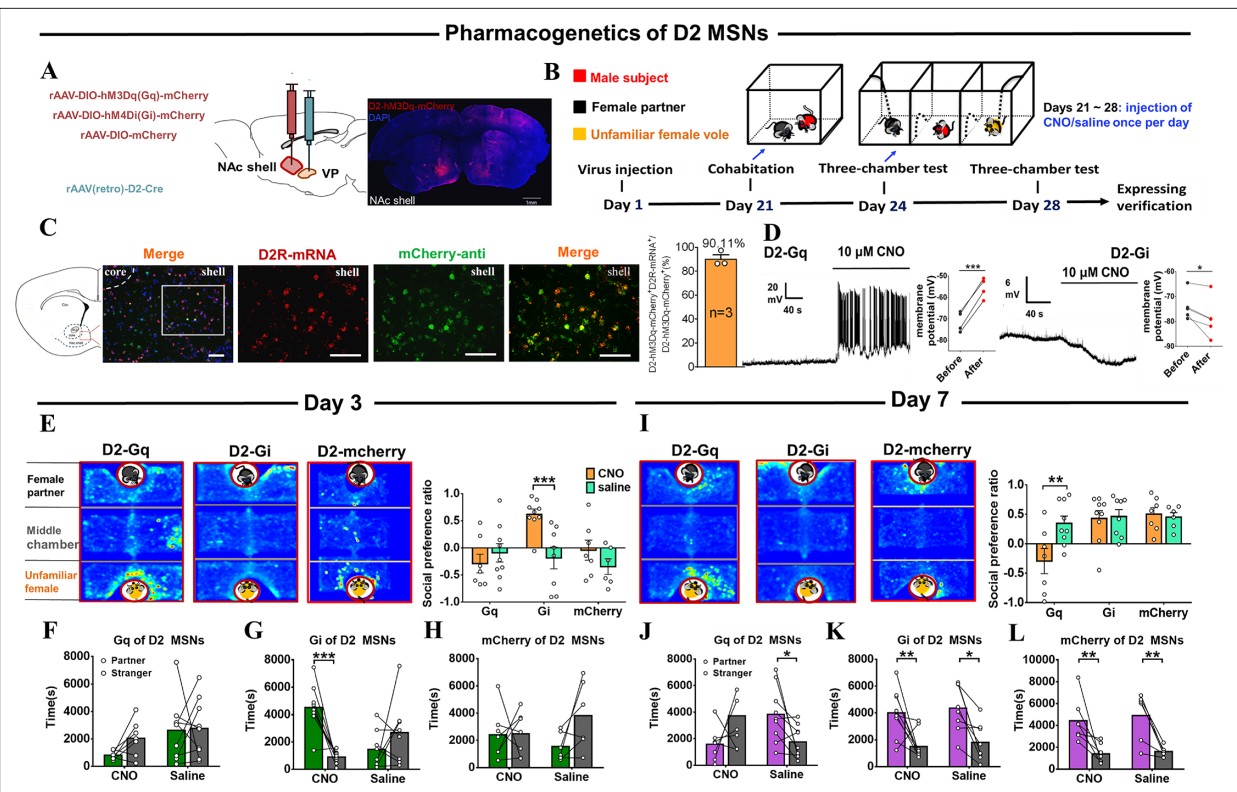

**Figure 6.** Effects of chemogenetic manipulation of NAc shell VP-projecting D2 MSNs on the formation of a partner preference. (**A**) Schematic of the chemogenetic viral strategy and injection sites as well as histology showing the expression of D2-hM3Dq-mcherry within the NAc shell. Scale bar: 1 mm. (**B**) Timeline of experiments. (**C**) Immunohistology image showing co-localization of hM3Dq-mCherry-anti expression (green), D2R-mRNA (red), and DAPI (blue) in the NAc shell. Scale bar: 100 μm. Statistical chart showing that D2-mRNA cells were relatively restricted to D2-hM3Dq-mCherry cells (n=3 voles). (**D**) Representative traces from a Gq-DREADD (left) neuron and Gi-DREADD (right) neuron after CNO bath. (**E**) Representative heatmaps of the partner preference test after 3 days of cohabitation. Left: Gq group. Middle: Gi group. Right: mcherry group. (**F–H**) Quantification of side-by-side times in partner preference tests after cohabitation for 3 days. (**I**) Representative heatmaps of the partner preference test after cohabitation for 7 days. Left: Gq group. Middle: Gi group. Right: mcherry group. (**J–L**) Quantification of side-by-side times in the partner preference test after cohabitation for 7 days. (D2-hM3Dq: CNO: n=7, saline: n=9; D2-hM4Di: CNO: n=9, saline: n=8; D2-mCherry: CNO: n=7, saline: n=6). Error bars = SEM. * represents p<0.05, ** represents p<0.01, and *** represents p<0.001. See *Figure 6—source data 1* for detailed statistics.

The online version of this article includes the following source data and figure supplement(s) for figure 6:

**Source data 1.** Raw data and statistical results of social preference ratio and side-by-side times in partner preference test after cohabitation in Gq group, Gi group, and mcherry group.

**Figure supplement 1.** The supplement Immunohistochemistry image of the D2 MSNs chemogenetics test.

**Figure supplement 1—source data 1.** Counts of mCherry-anti (green) positive neurons, D2R-mRNA (red) positive neurons, and co-expressed cells in hM3Dq, hM4Di, and mCherry group.

**Figure supplement 2.** The supplement FISH image of the D2-mCherry MSNs in the NAc shell.

**Figure supplement 2—source data 1.** Counts of D2-mCherry (green) positive neurons, D1R-mRNA (red) positive neurons, and co-expressed cells.

preferences. Nevertheless, CNO-treated voles with VP-projecting D2 MSNs Gq-DREADD virus infection spent less time with their partners than with unknown females in the partner preference test; moreover, such activation of VP-projecting D2 MSNs also blocked partner preference after 7 days of cohabitation (*Figure 6I and J*; 6I, Two-way ANOVA: group ×treatment: F (2, 40)=3.798, p=0.031; group: F (2, 40)=6.797, p=0.003; treatment: F (1, 40)=3.375, p=0.074; Gi_Saline vs Gi_CNO, p=0.861; Gq_Saline vs Gq_CNO, p=0.002; mCherry_Saline vs mCherry_CNO, p=0.805. 6 J, Kruskal-Wallis: H=8.083, p=0.044; CNO_Partner vs CNO_Stranger, p=0.384; saline_Partner vs saline_Stranger, p=0.033). In control virus subjects, CNO had no detectable effects on the behavioral performance in tests (*Figure 6L*); (Two-Way Repeated Measures ANOVA: group ×treatment: F (1, 11)=0.043, p=0.839; group: F (1, 11)=0.306, p=0.591; treatment: F (1, 11)=25.166, p=0.0004; CNO_Partner vs CNO_Stranger, p=0.009; saline_Partner vs saline_Stranger, p=0.009).

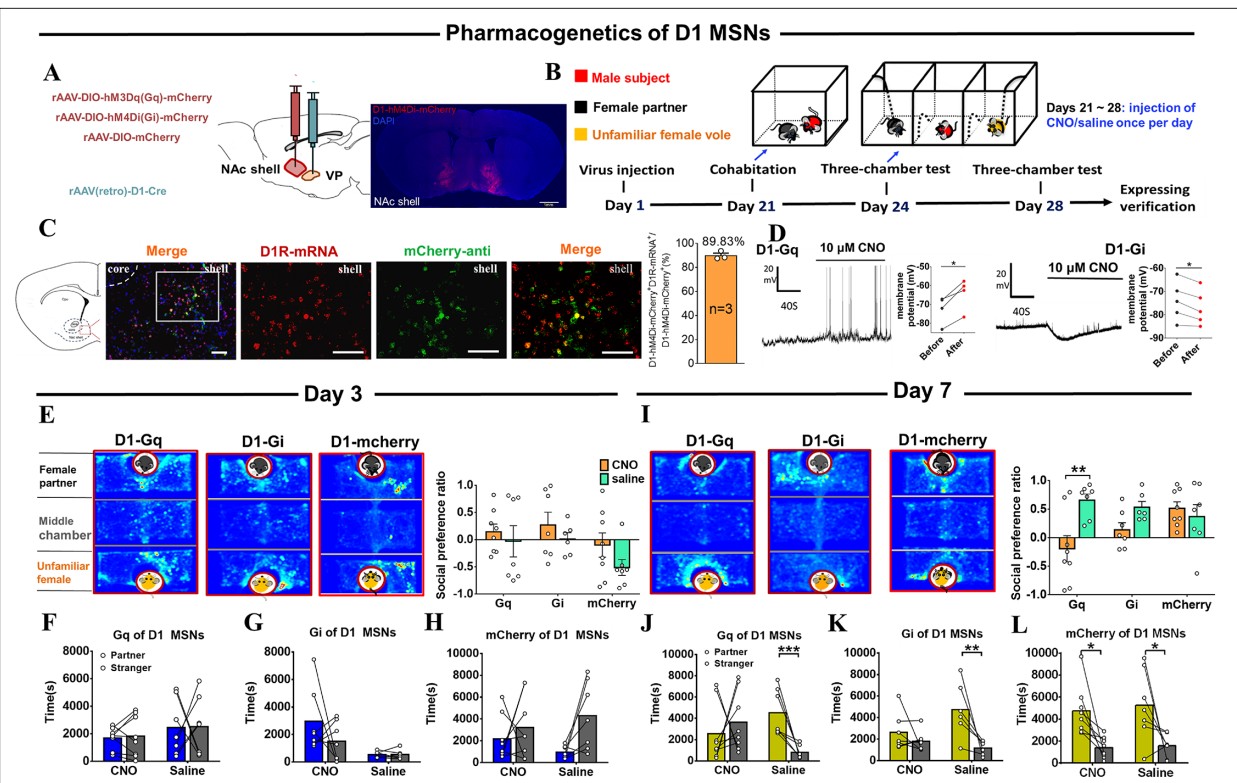

**Figure 7.** Effects of chemogenetic manipulation of NAc shell VP-projecting D1 MSNs on the formation of partner preference. (**A**) Schematic of chemogenetic viral strategy and injection sites as well as histology showing the expression of D1-hM4Di-mcherry within the NAc shell. Scale bar: 1 mm. (**B**) Timeline of experiments. (**C**) Immunohistology image showing co-localization of hM4Di-mCherry-anti expression (green), D1R-mRNA (red), and DAPI (blue) in the NAc shell. Scale bar: 100 μm. Statistical chart showing that D1-mRNA cells were relatively restricted to D1-hM4Di-mCherry cells (n=3 voles). (**D**) Representative traces of a Gq-DREADD (left) neuron and Gi-DREADD (right) neuron after CNO bath. (**E**) Representative heatmaps of the partner preference test after 3 days of cohabitation. Left: Gq group. Middle: Gi group. Right: mcherry group. (**F–H**) Quantification of side-by-side times in the partner preference test after 3 days of cohabitation. (**I**) Representative heatmaps of the partner preference test after 7 days of cohabitation. Left: Gq group. Middle: Gi group. Right: mcherry group. (**J–L**) Quantification of side-by-side times in the partner preference test after 7 days of cohabitation. (D1-hM3Dq: CNO: n=8, saline: n=7; D1-hM4Di: CNO: n=7, saline: n=6; D1-mCherry: CNO: n=8, saline: n=7). Error bars = SEM. * represents p<0.05, ** represents p<0.01, and *** represents p<0.001. See *Figure 7—source data 1* for detailed statistics.

The online version of this article includes the following source data and figure supplement(s) for figure 7:

**Source data 1.** Raw data and statistical results of social preference ratio and side-by-side times in partner preference test after cohabitation in Gq group, Gi group, and mcherry group.

**Figure supplement 1.** The supplement Immunohistochemistry image of the D1 MSNs chemogenetics test.

**Figure supplement 1—source data 1.** Counts of mCherry-anti (green) positive neurons, D1R-mRNA (red) positive neurons, and co-expressed cells in hM3Dq, hM4Di, and mCherry group.

**Figure supplement 2.** The supplement FISH image of the D1-mCherry MSNs in the NAc shell.

**Figure supplement 2—source data 1.** Counts of D1-mCherry (green) positive neurons, D2R-mRNA (red) positive neurons, and co-expressed cells.

**Figure supplement 3.** Neurobiological mechanism and circuit mechanism underlying the formation of a pair bond.

Based on the results of chemogenetic manipulations described above, it was concluded that the activation of D2 MSNs projecting to VP disrupted the formation of the pair bond, while its inhibition improved the formation of partner preference in mandarin voles.

The chemogenetic approach was similarly used to test whether VP-projecting D1 MSNs in NAc shell regulate partner preference. rAAV-DIO-hM3Dq-mCherry or rAAV-DIO-hM4Di-mCherry and rAAV (Retro)-D1-Cre virus were bilaterally injected into the NAc shell and VP, respectively, to inhibit VP-projecting D1 MSNs (*Figure 7A*). Similarly, male mandarin voles received daily intraperitoneal injections of CNO (1 mg/kg) or saline during cohabitation, and partner preference was tested on days 3 and 7. In the partner preference test, CNO-treated voles with VP-projecting D1 MSNs Gi-DREADD virus infection showed trends to reduce side-by-side contact with their partners compared to saline-treated

voles in the partner preference test after 7 days of cohabitation (*Figure 7E, G, I and K*; 7E, Two-way ANOVA: group ×treatment: F (2, 37)=0.186, p=0.831; group: F (2, 37)=2.729, p=0.078; treatment: F (1, 37)=2.839, p=0.100. 7 G, Kruskal-Wallis: H=9.597, p=0.022; CNO_Partner vs CNO_Stranger, p=0.087; saline_Partner vs saline_Stranger, p=0.821. 7I, Two-way ANOVA: group ×treatment: F (2, 37)=4.934, p=0.013; group: F (2, 37)=0.873, p=0.426; treatment: F (1, 37)=7.633, p=0.009; Gq_Saline vs Gq_CNO, p=0.001; Gi_Saline vs Gi_CNO, p=0.109; mCherry_Saline vs mCherry_CNO, *P*=0.522. 7 K, Two-Way Repeated Measures ANOVA: group ×treatment: F (1, 11)=4.281, p=0.063; group: F (1, 11)=1.513, p=0.244; treatment: F (1, 11)=11.027, p=0.007; CNO_Partner vs CNO_Stranger, p=0.377; saline_Partner vs saline_Stranger, p=0.004). In addition, CNO-treated voles with VP-projecting D1 MSNs Gq-DREADD virus infection showed a significant reduction in side-by-side time spent with their partner than saline-treated voles in the partner preference test after 7 days of cohabitation and showed no partner preference (*Figure 7E, F, I and J*) (7 F, Kruskal-Wallis: H=0.743, p=0.863. 7 J, Kruskal-Wallis: H=12.819, p=0.005; CNO_Partner vs CNO_Stranger, p=0.256; saline_Partner vs saline_Stranger, p=0.001). In control virus subjects, CNO had no significant effect on behavioral performance during testing (*Figure 7L*; Two-Way Repeated Measures ANOVA: group ×treatment: F (1, 13)=0.033, p=0.859; group: F (1, 13)=0.325, p=0.578; treatment: F (1, 13)=15.559, p=0.002; CNO_Partner vs CNO_Stranger, p=0.033; saline_Partner vs saline_Stranger, p=0.029). These results indicate that activation of the VP-projecting D1 MSNs impaired partner preference, but inhibition of these neurons did not produce significant effects on partner preference.

## Discussion

Although previous studies have demonstrated an association between the DA system and pair bond formation using pharmacological methods, the detailed neural mechanisms by which DA modulates partner preference remain elusive. By using AAV virus vectors and whole-cell patch-clamp recording, the present study found that, after 7 days of cohabitation, extracellular DA release in the NAc shell increased with social interaction; this increase was higher in response to a familiar partner than in response to an unfamiliar female. The observed increase in DA release resulted in alterations in the activities of D2 and D1 neurons in the NAc shell selectively in response to their partner after extended cohabitation. Furthermore, the electrophysiological properties of D1 and D2 neurons in the NAc shell were also altered by cohabitation, displaying neural plasticity. Finally, opposite effects of DA on D1 and D2 neurons are supported by circuit manipulation, namely activation of D2 MSNs projections to VP or inhibiting/activating D1 MSNs projections to VP inhibits pair bonding, while inhibition of D2 MSNs projections to VP promotes pair bonding. These results reveal a neurobiological mechanism underlying the attachment to a partner and circuit mechanism underlying the formation of a pair bond.

To validate the choice of 3 days of cohabitation as an indicative timepoint of the DA response 'before the pair bond'", an additional experiment was conducted to measure DA release in the NAc upon sniffing their partner and a stranger after 1 hr and 3 days of cohabitation. The results affirmed that 1 hr of cohabitation was not sufficient to form pair bonding. Fluorescence signals of the DA release in the NAc shell did not show differences upon sniffing their partner and a stranger after 1 hr and 3 days of cohabitation (*Figure 1—figure supplement 3*). After 3 days of cohabitation, males showed similar levels of DA release upon sniffing their partner and a stranger than males after 1 hr of cohabitation. Although according to partner preference tests, certain individuals preferred to stay with their partner after 3 days of cohabitation because of individual variation, the statistical result did not show significant preference for a certain partner. This result indicates that 3 days of cohabitation were not sufficient to form a stable pair bond at the behavioral and molecular levels.

In the present study, after 7 days of cohabitation, male mandarin voles showed more real-time DA release while they were sniffing their partner or had side-by-side contact with their partner than during the same behaviors directed toward an unknown female (*Figure 1*, *Figure 1—figure supplement 8A, B*). Previous studies corroborated the important role of DA in the formation of a pair bond (*Aragona et al., 2003*; *Wang et al., 2004*). Moreover, local injections of the DA agonist apomorphine into the NAc shell enhanced partner preference formation; microinjection of the DA antagonist haloperidol into the NAc shell could prevent the formation of mating-induced partner preferences (*Aragona et al., 2003*). These results are also in line with previous reports, where mating was found to increase DA levels in the NAc among both male and female prairie voles (*Aragona et al., 2003*; *Gingrich et al., 2000*). Similarly, the results of the present study are also consistent with a recent

study showing that upon sniffing and huddling with their partner, DA release was higher than during the same behaviors directed toward an unfamiliar vole (***Pierce et al., 2024***). In addition, the obtained results showed that extracellular DA levels upon sniffing their partner after 7 days of cohabitation were significantly higher compared to the extracellular DA after 3 days of cohabitation (***Figure 1—figure supplement 9B***). Furthermore, there was no significant difference in the fluorescence signals between same-sex strangers and cagemates after 3 or 7 days of cohabitation (***Figure 1—figure supplement 4***, ***Figure 2—figure supplement 2***, and ***Figure 3—figure supplement 2***). Changes in fluorescence signals were also not found during non-social behavioral bouts (***Figure 1—figure supplement 9A***). Combined with this experimental evidence, it is suggested that differences in the levels of DA release upon sniffing a partner and a stranger after 7 days of cohabitation may be due to the formation of a pair bond.

In addition, changes that were detected in the DA signal during side-by-side contact were much smaller than those detected upon sniffing (***Figure 1***, ***Figure 1—figure supplement 8***). It is possible that the DA level was significantly elevated during the initial encounter, thus DA increased strongly during initial examination (***Dai et al., 2022***). One prominent feature of the examination-associated DA response is its fast adaptation, which diminishes exponentially later on ***Dai et al., 2022***. If sniffing is considered as an appetitive goal-directed behavior, side-by-side alignment may be a consummatory behavior. DA release generally precedes the onset of consummatory behaviors; as this behavior lasts, DA release may decrease significantly (***Dai et al., 2022***). In addition, DA is critical for the appetitive drive for social interaction, but not for low-effort, unconditioned consummatory behaviors (***Pierce et al., 2024***). Side-by-side contact is a long-lasting behavior that happens in a quiescent state. These may be the reasons why, upon sniffing, the DA signal is stronger than that upon side-by-side alignment.

Furthermore, the present study found that D2 MSNs showed more enhanced suppression of activity after 7 days of cohabitation upon sniffing their partner compared with the activity after 3 days of cohabitation (***Figure 2—figure supplement 7B***). Similarly, D1 MSNs showed a significant difference in response to sniffing their partner after 7 days of cohabitation compared to the response after 3 days of cohabitation (***Figure 3—figure supplement 7B***). These results are supported by pharmacological research, showing that activation of D2R in the NAc shell of female prairie voles accelerates partner preferences without mating, and the blockage of D2R antagonizes this behavior (***Young and Wang, 2004***). In addition, selective aggression toward a stranger was stopped by blocking D1R in the NAc (***Resendez et al., 2012***). In the present study, increasing DA release in the NAc, decreasing calcium activity of D2 MSNs, and increasing calcium activity of D1 MSNs during social interaction with their partner after pair bond formation may be one mechanism underlying the preference for the partner after the formation of a pair bond.

In addition, electrophysiological properties and synaptic plasticity of MSNs expressing D1R and D2R after pair bonding were characterized for the first time. The NAc is assumed to play a role in emotion- and motivation-related learning and memory (***Cardinal and Everitt, 2004***; ***Kelley, 2004***). DA in the NAc is assumed to promote pair bonding by facilitating the association between sensory cues of the partner and the reward of sexual behavior (***Walum and Young, 2018***). The sexual experience gained during cohabitation and pair bond formation may alter the electrophysiological properties of D1 and D2 MSNs. In a previous study, both D1R and D2R could specifically regulate the formation of partner preference (***Young et al., 2011***). Activation of D1R and D2R has different effects on the cyclic AMP (cAMP) signaling pathway, which has been proved to underlie the specific regulation of pair bond formation by different types of DA receptors (***Aragona and Wang, 2007***). The physiological and direct cellular consequences of DA action on MSNs are the positive or negative effects of cAMP production, mediated by D1R or D2R, respectively (***Lobo and Nestler, 2011***). Although DR in the NAc shell is known to have direct connections with pair bond formation according to studies using animal models (***Aragona et al., 2006***; ***Young and Wang, 2004***), experimental evidence supporting a link between functional NAc shell MSNs and partner preference is still lacking. Although one recent study also found that pair bonding increases the amplitude of electrically induced EPSCs (***Borie et al., 2022***), the present study provides direct experimental evidence that the formation of partner preference induces different changes in both the neuronal activities and synaptic transmission of D1/D2 MSNs in male mandarin voles.

The present study found that the frequencies of sEPSCs and sIPSCs were significantly enhanced after the formation of a pair bond in NAc shell D2 MSNs. The excitatory/inhibitory balance of D2

MSNs was enhanced after cohabitation. These results are not consistent with the findings from fiber photometry of calcium signals. One study showed that NAc D2 MSNs were linked to both 'liking' (food consumption) and 'wanting' (food approach) but with opposing actions. The high D2 MSNs activity signaled 'wanting', and the low D2 MSNs activity enhanced 'liking'. D2 MSNs face a tradeoff between increasing 'wanting' by being more active or allowing 'liking' by remaining silent (*Guillaumin et al., 2023*). Therefore, the increase in frequencies of sEPSC and sIPSC in D2 MSNs may reflect two processes, liking and wanting differently and respectively. We suggest that hedonia and motivation might influence D2 MSNs activity during cohabitation and contribute to the processing of pair bond formation in a more dynamic and complex manner than previously expected. The difference may also be related to the interactions between neurotransmitters such as DA, glutamate, and gamma-aminobutyric acid (GABA) in the basal ganglia (*Gerfen and Surmeier, 2011*; *Kalivas et al., 2009*; *Meyer-Lindenberg et al., 2011*; *Pistillo et al., 2015*; *Shamay-Tsoory and Abu-Akel, 2016*). DA modulates neurotransmitter release at both GABAergic and glutamatergic striatal synapses (*Lovinger, 2010*). It has been reported that the mEPSC amplitude is related to partner preference in prairie voles, but no difference in mEPSC frequency was detected between the NAc of voles and rats (*Willett et al., 2018*), which were not consistent with present results. The possible reasons for this variance are differences in brain regions (core and shell), and the types of MSNs were further distinguished in this research, which was neglected by previous studies. The frequency of mEPSC in overall MSNs may produce no changes, given that a decreased frequency of sEPSC in D1 MSNs and an increased frequency of sEPSC in D2 MSNs were observed. Further, the formation of a pair bond increased the D2 MSNs excitation/inhibition balance and amplified the D2 MSNs output in the present study, which would affect information processing in the NAc. Moreover, the frequencies of sEPSC and sIPSC were significantly reduced in the NAc shell D1 MSNs after pair bonding, whereas the intrinsic excitability increased after cohabitation with females. The bidirectional modifications (reduced synaptic inputs vs. increased excitability) observed in D1 MSNs might result from homeostatic regulation. The overall synaptic transmission may produce no net changes, given that reductions in both excitatory and inhibitory synaptic transmission of D1 MSNs were observed. Also, increases in the intrinsic excitability of D1 MSNs would result in an overall excitation gain on D1 MSNs. There are several explanations for the result that the same concentration of DA applied in a bath with slices from the cohabitation group did not change the frequency of D2/D1 sEPSC or the amplitude of sEPSCs. During cohabitation, the NAc shell DA concentration increased with mating behaviors (*Aragona et al., 2003*). The DA receptor has been activated, resulting in reduced DA receptor sensitivity to DA in the cohabitation group. In addition, DA exerts its role by binding to G protein-coupled receptors (i.e. D1R and D2R) that produce positive or negative effects via cAMP (*Lobo and Nestler, 2011*). In addition to DA, other neurotransmitters, such as glutamic acid, also produce effects via the same pathway. It remains unclear how these neurotransmitters interact in the MSNs to mediate activities. How other neural circuits (basolateral amygdala-NAc) jointly change the neuronal activity of MSNs after cohabitation has not yet been explored, which suggests a promising avenue for exploration in the future. Further studies of the distinct intracellular or synaptic mechanisms of both types of MSNs and their dynamics at various times during pair bond formation are necessary to better understand the cell-type-specific functions and physiological consequences of pair bonding.

Moreover, the mesolimbic DA system regulates pair bond formation through the NAc-VP circuit (*Numan and Young, 2016*). The D1/D2 MSNs from the NAc shell equally innervate the VP through inhibitory projections (*Kupchik et al., 2015*). Combined with the findings in changes in DA release, calcium signals, and electrophysiological properties, it can be speculated that the activation of VP-projecting D2 MSNs caused by the DA release and the negative regulation loop of VP-projecting D1 MSNs in the NAc shell may result in higher excitability of VP and the promotion of pair bond formation. Further, the effects of DA on NAc seem to promote synaptic plasticity, enabling mating partners to continuously activate the NAc-VP attraction circuit, finally leading to lasting social attraction and bonding.

In the present study, DREADD's approaches were used to inhibit or excite NAc MSNs to VP projection and it was found that D1 and D2 NAc MSNs projecting to VP play different roles in the formation of a pair bond. Chemogenetic inhibition of VP-projecting D2 MSNs promoted partner preference formation, while activation of VP-projecting D2 MSNs inhibited it (*Figure 6*). Chemogenetic activation of D2 MSNs produced the opposite effect of DA on the D2 MSNs on partner preference, while

inhibition of these neurons produced the same effect of DA on D2 MSNs. DA binding with D2R is coupled with Gi and produces an inhibitory effect (*Lobo and Nestler, 2011*). It is generally assumed that the activation of D2R produces aversive and negative reinforcement. These results were consistent with the reduced activity of D2 MSNs upon sniffing their partner in the fiber photometry test and the increased frequency and amplitude of sIPSC in the present study. Our results also align with other previous studies, which showed that chemogenetic inhibition of NAc D2 MSNs is sufficient to enhance reward-oriented motivation in motivational tasks (*Carvalho Poyraz et al., 2016*; *Gallo et al., 2018*). Inhibition of D2 MSNs during self-administration enhanced the response and motivation to obtain cocaine (*Bock et al., 2013*). This also suggests that the mechanism underlying attachment to a partner and drug addiction is similar.

Besides, in the present study, the formation of partner preference was inhibited following activation or inhibition of VP-projecting D1 MSNs, which is not consistent with conventional understanding of prairie vole behavior. Alternatively, DA binding with D1R is coupled with Gs and produces an excitatory effect (*Lobo and Nestler, 2011*), while the activation of D1R produces reward and positive reinforcement (*Hikida et al., 2010*; *Kwak and Jung, 2019*; *Tai et al., 2012*). For example, activation of D1 MSNs enhances the cocaine-induced conditioned place preference (*Lobo et al., 2010*). In addition, D1R activation by DA promotes D1 MSNs activation, which in turn promotes reinforcement. However, a recent study found that NAc-ventral mesencephalon D1 MSNs promote reward and positive reinforcement learning; in contrast, NAc-VP D1 MSNs led to aversion and negative reinforcement learning (*Liu et al., 2022*). It is consistent with our results that activation of the NAc-VP D1 MSNs pathway reduced time spent side-by-side and impaired partner preference after 7 days of cohabitation. In contrast to the inhibition of D2 MSNs, we found that inhibition of D1 MSNs did not elicit corresponding increases in partner preference. One possible explanation is that almost all D1 MSNs projecting to the VTA/substantia nigra (SN) send collaterals to the VP (*Pardo-Garcia et al., 2019*). For example, optogenetically stimulating VP axons may inadvertently cause effects in the VTA/SN through the antidromic activation of axon collaterals (*Yizhar et al., 2011*). Therefore, chemogenetic inhibition of D1 MSNs may also inhibit DA neurons in the VTA, subsequently inhibiting the formation of a pair bond.

The DA and different types of DA receptors in the NAc may play different roles in the regulation of pair bond formation and maintenance. The chemogenetic manipulation has revealed that VP-projecting D2 MSNs are necessary and more important in pair bond formation compared to VP-projecting D1 MSNs. It is consistent with previous pharmacological experiments that blocking D2R with its specific antagonist, while D1R was not blocked, can prevent the formation of a pair bond in prairie voles (*Gingrich et al., 2000*). This indicates that D2R is crucial for the initial formation of the pair bond. D2R is involved in the reward aspects related to mating. In female prairie voles, D2R in the NAc is important for partner preference formation. The activation of D2R may help to condition the brain to assign a positive valence to the partner's cues during mating, facilitating the development of a preference for a particular mate. In addition, the cohabitation caused the DA release, and the high-affinity Gi-coupled D2R was activated first, which inhibited D2 MSNs activity and promoted the pair bond formation. And then, after 7 days of cohabitation, the pair bonding was already established. The significantly increased release of DA significantly activated Gs-coupled D1R with the low affinity to DA, which increased D1 MSNs activity and maintained the formation of partner preference. While D1R is also present and involved in the overall process, its role in the initial formation of the pair bond is not as dominant as D2R (*Aragona et al., 2006*). However, it still participates in the neurobiological processes related to pair bond formation. For example, in male mandarin voles, after 7 days of cohabitation with females, D1R activity in the NAc shell was affected during pair bond formation. The extracellular DA concentration was higher when sniffing their partner compared to a stranger, and this increase in DA release led to an increase in D1R activity in the NAc shell. In prairie voles, DA D1 receptors seem to be essential for pair bond maintenance. Neonatal treatment with D1 agonists can impair partner preference formation later in life, suggesting an organizational role for D1 in maintaining the bond (*Aragona et al., 2006*). In pair-bonded male prairie voles, D1R is involved in inducing aggressive behavior toward strangers, which helps to maintain the pair bond by protecting it from potential rivals. In the NAc shell, D1 agonist decreases the latency to attack same-sex conspecifics, while D1 antagonism increases it (*Aragona et al., 2006*). In summary, D2R is more crucial for pair bond formation, being involved in reward association and necessary for the initial development of the bond. D1R, on

the other hand, is more important for pair bond maintenance, being involved in aggression and mate guarding behaviors and having an organizational role in maintaining the bond over time. We therefore suggest that D2 MSNs are more predominantly involved in the formation of a pair bond compared with D1 MSNs.

However, certain limitations of the present study should not be ignored. Firstly, in the present study, it was observed that about 20% of D1 promoter MSNs were labeled by D2-mRNA and about 14% D2 promoter MSNs were labeled by D1-mRNA vice (*Figure 2—figure supplement 8*, *Figure 3—figure supplement 8*, *Figure 6—figure supplement 2*, *Figure 7—figure supplement 2*). One reason for this difference may be that a small part of MSNs can co-express both D1R and D2R in the NAc shell (*Valjent et al., 2009*). Another reason may be the non-specific expression of the virus. This means that a small group of MSNs was misclassified and miscounted. Although the effect of co-expressed neurons cannot be completely eliminated, the obtained results still reflect the changes in activity and physiological function of the vast majority of MSNs after the formation of a pair bond. In addition to DA, a variety of neurotransmitters or modulators are also involved in the formation of a pair bond, such as oxytocin, glutamate, and GABA. In the future, it is of great importance to study the synergistic regulation of pair bonding by multiple transmitters and systems in the NAc shell. Secondly, it is important to further examine whether other neural circuits also influence the formation of partner preference, such as basolateral amygdala projections to NAc. Moreover, in general, natal philopatry among mammals is female-biased in the wild (*Brody and Armitage, 1985*; *Greenwood, 1983*; *Ims, 1990*; *Solomon and Jacquot, 2002*). Social mammals are rarely characterized by exclusively male natal philopatry (*Solomon and Jacquot, 2002*). Males often disperse from the natal area to a new place. Thus, male rodents may play a dominant role in the formation and maintenance of mating relationships. This is a reason why we investigate pair bonding in males firstly. Certainly, female mate selection, as well as sexual receptivity or refusal through olfactory cues from males, thereby affects the formation and maintenance of pair bonding (*Hoglen and Manoli, 2022*). This is also the reason why we should focus on the mechanisms underlying pair bonding formation in females in future research. Despite these limitations, the obtained results reveal the neurobiological mechanism underlying attachment to a partner as well as the circuit mechanism underlying the formation of a pair bond (*Figure 7—figure supplement 3*).

## Materials and methods

**Key resources table**

| Reagent type (species) or resource | Designation | Source or reference | Identifiers | Additional information |
|---|---|---|---|---|
| Recombinant DNA reagent | rAAV-CAG-dLight1.1-WPRE-hGH_polyA | BrainVTA | PT-1138 | Serotype titer 2/9, $5.00×10^{12}$ vg/ml |
| Recombinant DNA reagent | rAAV-D1-GCaMp6m-WPRE-hGH_polyA | BrainVTA | PT-2749 | Serotype titer 2/9, $2.55×10^{12}$ vg/ml |
| Recombinant DNA reagent | rAAV-D2-GCaMp6m-WPRE-hGH_polyA | BrainVTA | PT-2750 | Serotype titer 2/9, $2.69×10^{12}$ vg/ml |
| Recombinant DNA reagent | rAAV-D1-EGFP-WPRE-bGH_polyA | BrainVTA | PT-0214 | Serotype titer 2/9, $2.78×10^{12}$ vg/ml |
| Recombinant DNA reagent | rAAV-D2-EGFP-WPRE-hGH_polyA | BrainVTA | PT-3245 | Serotype titer 2/9, $2.26×10^{12}$ vg/ml |
| Recombinant DNA reagent | rAAV-D1-mCherry-WPRE-bGH_polyA | BrainVTA | PT-0757 | Serotype titer 2/9, $3.66×10^{12}$ vg/ml |
| Recombinant DNA reagent | rAAV-D2-mCherry-WPRE-bGH_polyA | BrainVTA | PT-0367 | Serotype titer 2/9, $2.44×10^{12}$ vg/ml |
| Recombinant DNA reagent | rAAV-D1-CRE-WPRE-hGH_polyA | BrainVTA | PT-1217 | Serotype titer 2 /R, $4.72×10^{12}$ vg/ml |
| Recombinant DNA reagent | rAAV-D2-CRE-WPRE-bGH_polyA | BrainVTA | PT-0571 | Serotype titer 2 /R, $5.00×10^{12}$ vg/ml |

*Continued on next page*

*Continued*

| Reagent type (species) or resource | Designation | Source or reference | Identifiers | Additional information |
|---|---|---|---|---|
| Recombinant DNA reagent | rAAV-CAG-EGFP | BrainVTA | PT-0305 | Serotype titer 2/9, $5.28 \times 10^{12}$ vg/ml |
| Recombinant DNA reagent | rAAV-EF1α-DIO-hM3D(Gq)-mCherry | BrainVTA | PT-0042 | Serotype titer 2/9, $2.70 \times 10^{12}$ vg/ml |
| Recombinant DNA reagent | rAAV-EF1α-DIO-hM4D(Gi)-mCherry | BrainVTA | PT-0043 | Serotype titer 2/9, $5.53 \times 10^{12}$ vg/ml |
| Recombinant DNA reagent | rAAV- EF1α-DIO-mCherry | BrainVTA | PT-0013 | Serotype titer 2/9, $5.19 \times 10^{12}$ vg/ml |
| Antibody | Anti-mCherry | abcam | ab183628 | 1:300 |
| Antibody | anti-rabbit goat conjugated with Alexa Fluor 488 | JacksonImmuno | 111-545-003 | 1: 500 |
| Recombinant DNA reagent | RNAscope Probe-Mo-Drd1-C3 | Advanced Cell Diagnostics | #588161-C3 | |
| Recombinant DNA reagent | RNAscope Probe-Mo-Drd2-C2 | Advanced Cell Diagnostics | #534471-C2 | |
| Chemical compound, drug | RNAscope Multiplex Fluorescent Reagent Kit v2 | Advanced Cell Diagnostics | 323100 | |
| Chemical compound, drug | Clozapine N-oxide dihydrochloride | BrainVTA | CNO-02 | |
| Chemical compound, drug | 17-β-Estradiol-3-Benzoate | Sigma | E8875 | |
| Chemical compound, drug | Dopamine | Sigma | H8502 | |
| Chemical compound, drug | picrotoxinin | Sigma | P8390 | |
| Chemical compound, drug | CNQX | Sigma | 504914 | |
| Chemical compound, drug | D-AP5 | Sigma | 165304 | |
| Software, algorithm | MATLAB | MathWorks | RRID:SCR_001622 | |
| Software, algorithm | JWatcher | http://www.jwatcher.ucla.edu/ | RRID:SCR_017595 | |
| Software, algorithm | SPSS | IBM | RRID:SCR_002865 | |

## Animals

Subjects used in the present study were the F2 generation of mandarin voles bred in the laboratory. The voles were weaned 21 days after birth and lived in same-sex colonies in different polycarbonate cages (44×22 × 16 cm) and were housed in a 22–24 °C, 12 hr light/dark cycle with food and water ad libitum. The voles used in the experiment were ~70–90 days old. All laboratory procedures were conducted in accordance with the Guidelines for the Care and Use of Laboratory Animals in China and the regulations of the Animal Care and Use Committee of Shaanxi Normal University. This study protocol was reviewed and approved by the Academic Committee of Shaanxi Normal University, Special Committee of Scientific Ethics, Approval No. 202112005.

## Stereotaxic surgery and virus infusions

Male mandarin voles were anesthetized with 1.5–3.0% isoflurane inhalant gas (R5-22-10, RWD, China) and placed in stereotaxic instruments (68045, RWD, China). The brain coordinates of virus injection were as follows: NAc shell: AP:+1.5, ML:±0.99, DV: −4.2; VP: AP:+0.8, ML:±1.2, DV: −4.9. Next, a 10 µL Hamilton microsyringe (7635–01, HAMILTON) was used to inject the virus using a microsyringe pump (KDS legato 130, RWD, China) at a rate of 80 nl/min. After the injection, the needle was left in place for 10 min and was then slowly removed to prevent virus leakage. For fiber photometry recording, voles were mounted on a stereotaxic apparatus and a fiber optical cannula (diameter 2.5 mm, NA 0.37, depth 6 mm, RWD, China) was implanted into the site ~0.15 mm above the NAc shell 10 days after virus injection. Only voles with the correct locations of virus expression and optical fibers were used for further analyses.

Firstly, real-time release of DA and activity of two types of MSNs in the NAc shell upon sniffing their partner, an unknown female, or an object on days 3 and 7 of cohabitation were measured using an in

vivo fiber photometry system. Male mandarin voles were injected with rAAV-CAG-dLight1.1 (400 nl), rAAV-D1-GCaMP6m (400 nl), rAAV-D2-GCaMP6m (400 nl), rAAV-CAG-EGFP (300 nl), rAAV-D1-EGFP (400 nl), and rAAV-D2-EGFP (400 nl) into the NAc shell. The viruses used in the present study were obtained from brainVTA company and listed in the Key resources table. The non-coding promoter sequence of the mouse D1R/D2R gene was predicted and amplified for validation by the company. The sequence was then constructed and packaged by the rAAV virus vector by brainVTA company. The detailed sequence information can be obtained from brainVTA company.

Alterations in the neural excitability and synaptic plasticity of NAc shell D1/D2 MSNs after cohabitation were then examined using whole cell patch-clamp recordings. Animals were divided into sexually naive and cohabitation groups. Voles were injected with rAAV-D1-mCherry (400 nl) or rAAV-D2-mCherry (400 nl) into the NAc shell to identify D1/D2 MSNs. Twenty-one days after virus injection, the males were cohabitated with females for 7 days.

Finally, a chemogenetic approach was used to activate or inhibit D1/D2 MSNs projecting to the VP on days 3 and 7 of cohabitation to disclose different roles of these two types of projections in the formation of a pair bond. rAAV-EF1α-DIO-hM3Dq(Gq)–mCherry (400 nl), rAAV–EF1α–DIO–hM4Di(Gi)-mCherry (400 nl), or rAAV-EF1α-DIO-mCherry (400 nl) were bilaterally injected into the NAc shell, and rAAV (Retro)-D1-Cre (400 nl) and rAAV (Retro)-D2-Cre (400 nl) were injected into the VP. The D1/D2 promoter sequences originated from mice. Subjects were divided into Gq, Gi, and mCherry groups. After cohabitation, the partner preference test was conducted on days 3 and 7 days of cohabitation.

All female voles were ovariectomized and primed through subcutaneous administration of estradiol benzoate (17-β-Estradiol-3-Benzoate, Sigma, 2 µg dissolved in sesame oil starting 3 days prior to the experiments; *Borie et al., 2022*).

## Fiber photometry

The fiber photometry system (ThinkerTech, Nanjing, China) was used as previously described (*Li et al., 2021*). Briefly, the emission light from modulated blue 480 LED (50 mW) was reflected with a dichroic mirror and then delivered to the brain by an optical objective lens coupled with an optical commutator, which excited the D1/D2-GCaMP6m or the DA sensor located in the NAc shell. The excitation light was passed through another band pass filter, into a CMOS detector (Thorlabs, Inc; DCC3240M), and finally recorded by LabVIEW software (TDMSViewer; ThinkerTech, Nanjing, China). Fourteen days after virus injection, male voles were cohabited with females. Previous results confirmed that male mandarin voles show significant preference to co-housed partners after 5–7 days of cohabitation (*Feng et al., 2021*); therefore, the real-time release of DA and the activity of D1/D2 MSNs in the NAc shell were observed on days 3 and 7 days of cohabitation (partner: DA sensor: day 3, n=7; day 7, n=7. D2-GCaMP6m: day 3, n=7; day 7, n=7. D1-GCaMP6m: day 3, n=7; day 7, n=7; same sex cagemate: DA sensor: day 3, n=6; day 7, n=6. D2-GCaMP6m: day 3, n=6; day 7, n=6. D1-GCaMP6m: day 3, n=6; day 7, n=6).

After 3 and 7 days of cohabitation, subjects were anesthetized with isoflurane and connected to a multi-mode optic fiber patch cord (ThinkerTech, Nanjing, China; NA: 0.37, OD: 200 µm) connected to a fiber photometry apparatus. The cage for social interaction is an open field (44×22 × 16 cm) in which subjects were exposed to their partner or a stranger. Males were exposed to partner or an unfamiliar female (each exposure lasted for 30 min) in random order in a clean social interaction cage. The changes in fluorescence signals during these social interactions with their partner, an unfamiliar vole of the opposite sex, or an object (Rubik's Cube) were collected and digitalized by CamFiberPhotometry software (ThinkerTech). To rule out that the difference in fluorescence signals was caused by the difference in virus expression at different time points, we used the same experimental strategy in new male mandarin voles and measured the fluorescence signal changes upon eating carrot after 3 and 7 days of cohabitation (The male mandarin voles were fasted for four hours before the test.). Fiber photometry was performed to measure the release of DA and the activity of D1/D2 MSNs in the NAc shell during social interaction with a partner or a stranger, including sniffing and side-by-side contact (The side-by-side contact behavior is defined as significant physical contact with a social vole and huddling in a quiescent state.). In the same-sex cagemate group, male mandarin voles were cohabited with a same-sex vole for 3 and 7 days. Males were then exposed to their same-sex cagemate or an unfamiliar male (each exposure lasted for 30 min and was conducted in random order) in a clean social interaction cage. The changes in fluorescence signals upon sniffing the same-sex cagemate, an

unfamiliar male vole, or an object were collected and digitalized by CamFiberPhotometry software (ThinkerTech).

All data were analyzed in MatLab 2019a; the ΔF/F values represent the release of DA and activity of D1/D2 MSNs during sniffing and side-by-side contact. The baseline ($F_0$) for all events and behaviors was the averaged fluorescence signal of –10–0 s prior to exposure to the stimulated voles. The fluorescence signals after the start of sniffing and side-by-side contact are recorded as F. These fluorescence signals during the entire 30-min session were recorded and analyzed. The fluorescence change (ΔF/F) values were calculated as $(F – F_0) / F_0$. Then, the time window was determined based on the duration of a behavioral bout. To estimate the calcium response, the average (ΔF/F) was calculated during the 4 s time window following the beginning of sniffing and side-by-side contact. Otherwise, the data points where the sniffing bout occurred within 4 s of a prior sniffing bout were excluded from data analysis. The sniffing and side-by-side contact that occurred in a fiber photometry experiment were examined using jwatcher, a behavior event scoring software (https://www.jwatcher.ucla.edu/, Dan Blumstein's Lab & the late Christopher Evans' lab, Sydney).

## Slice preparation and whole-cell patch clamp recording

Four weeks after virus injection, voles were anesthetized with 1.5–3.0% isoflurane inhalant gas and the brains were harvested into oxygenated artificial cerebrospinal fluid (ACSF) containing 125 mM sodium chloride (NaCl), 2.5 mM potassium chloride (KCl), 25 mM glucose, 25 mM sodium hydrogen carbonate ($NaHCO_3$), 1.25 mM monosodium phosphate ($NaH_2PO_4$), 2 mM calcium chloride ($CaCl_2$), and 1 mM magnesium chloride ($MgCl_2$), gassed with 5% $CO_2$ /95%$O_2$. Sagittal NAc slices (300 μm) were cut on a vibratome (VT 1200 S, Leica, Germany) with oxygenated ACSF (32–34 °C), incubated at 32 °C for 30 min, followed by 1 h of incubation at room temperature before recording. Voltage- and current-clamp whole-cell recordings were performed using standard techniques at room temperature. Electrodes were pulled from 1.5 mm borosilicate-glass pipettes on a P-97 puller (Sutter Instruments).

Whole-cell patch-clamp recordings were obtained with a Multiclamp 700B amplifier, digitized at 10 kHz using a Digidata 1440 A acquisition system with Clampex 10.2, and analyzed with pClamp 10.5 software (Molecular Devices). Only cells that maintained a stable access resistance (<30 MΩ) throughout the entire recording were analyzed. mCherry MSNs in the NAc shell were identified under IR-DIC optics and fluorescence microscopy. sIPSCs (D2 MSNs: Cohabitation, n=14 cells from four voles; Control, n=14 cells from four voles) (D1 MSNs: Cohabitation, n=16 cells from four voles; Control, n=10 cells from three voles) were isolated by adding D-AP5 (50 μM, Sigma), and CNQX (20 μM, Sigma) in ACSF while recording at a holding potential of 0 mV. sEPSCs (D2 MSNs: Cohabitation, n=11 cells from four voles; Control, n=11 cells from four voles) (D1 MSNs, Cohabitation: n=19 cells from five voles; Control, n=16 cells from four voles) were isolated by adding PTX (100 μM, Sigma) in ACSF while recording at a holding potential of –70 mV. For evoked glutamatergic and GABAergic PSC recordings, a concentric electrode was used to stimulate axon terminal input to the NAc shell. Evoked EPSCs were isolated by voltage-clamping neurons at the reversal potential of the inhibition (–70 mV), whereas evoked IPSCs were recorded at the reversal potential of the excitation (0 mV). A total of 20 recording events with intervals of 30 s at each holding potential were used for analysis. The E-I ratio was calculated from baseline subtracted traces as the average EPSC amplitude divided by the average IPSC amplitude. All data were normalized to the mean of the control E-I ratio (D2 MSNs: Cohabitation, n=7 cells from four voles; Control, n=5 cells from three voles) (D1 MSNs: Cohabitation, n=9 cells from four voles; Control, n=5 cells from three voles). For the bath-applied DA test, sEPSCs were isolated by adding PTX (100 μM) in ACSF while recording at a holding potential of –70 mV. After 10 min of recording, slices were perfused with 5 μM DA (Sigma). The total recording time for each cell was 40 min and the last 10 min of the recording were assessed (D2 MSNs: Cohabitation, n=8 cells from four voles; Control, n=8 cells from three voles; D1 MSNs: Cohabitation, n=10 cells from five voles; Control, n=10 cells from four voles). Electrode resistance was ~5–7 MΩ when filled with internal solution consisting of 130 mM $CsMeSO_3$, 8 mM CsCl, 1 mM MgCl2, 0.3 mM EGTA, 10 mM HEPES, 4 mM ATP (magnesium salt), 0.3 mM GTP (sodium salt), and 10 mM phosphocreatine (pH 7.4, 300 mOsm).

To assess the neuronal action potential, MSNs were stimulated with step-current pulses (1000ms, ranging from 0 to 250 pA) by adding PTX (100 μM, Sigma), CNQX (20 μM, Sigma), and D-AP5 (50 μM, Sigma). (D2 MSNs: Cohabitation, n=11 cells from four voles; Control, n=13 cells from six voles) (D1

MSNs: Cohabitation, n=8 cells from three voles; Control, n=8 cells from four voles). To determine the activation and inhibition effect of CNO (BrainVTA, CNO-02) in Gi or Gq virus-infected neurons, spontaneous firing of action potentials in the cell was recorded in the current-clamp mode. After 3 min of baseline recording, NAc slices from mice with injection of Gq and Gi viruses were perfused with 10 µM CNO for 7 min. The total recording time for each cell was 10 min. Electrode resistance was ~5–7 MΩ when filled with internal solution consisting of: 130 mM K-Gluconate, 5 mM NaCl, 10 mM HEPES, 0.5 mM EGTA, 2 mM Mg-ATP, and 0.3 mM Na-GTP (pH 7.3, 280 mOsm).

## Chemogenetics

Three weeks after virus injection, male mandarin voles were cohabited with females. For the D1-mCherry-hM3Dq/hM4Di group (hM3Dq: CNO, n=8; saline, n=7. hM4Di: CNO, n=7; saline, n=6) or the D2-mCherry-hM3Dq/hM4Di group (hM3Dq: CNO, n=7; saline, n=9. hM4Di: CNO, n=9; saline, n=8), male voles were injected with CNO (1 mg/kg, i.p. injection) or saline once per day during the 7 days cohabitation period. On days 3 and 7 of cohabitation, a partner preference test (3 h) was conducted 3 h after CNO injection. For the D1-mCherry group (CNO: n=8; saline: n=7) or the D2-mCherry group (CNO: n=7; saline: n=6), the same test procedure was used.

## Partner preference test

Partner preference formation is a reliable indicator of pair bonding and is characterized by selective contact and mating with partners rather than with strangers. The three-chamber partner preference test was first developed in Dr Sue Carter's lab and has since been adopted by many other laboratories (*Young et al., 2011*). The apparatus consists of a three-chamber (60×40 × 20 cm) arena where the middle chamber is connected to two identical chambers. Partner and unfamiliar voles of the opposite sex were placed into the two chambers on opposite sides. Before testing, subjects were adapted to the test arena for 30 min, and partner and opposite sex voles were also habituated for 10 min. During the 3 hr of partner preference test, two stimulus animals were firstly confined to their own chambers; then, the subjects were placed into the middle chamber and allowed to move freely. Real-time side-by-side contact behaviors were videotaped and quantified by smart 3.0. A longer duration of side-by-side contact spent with partners (but not unfamiliar opposite-sex female voles) indicates successful formation of partner preferences. The social preference ratio was calculated as follows: (time spent on partner side - time spent on stranger side) / (time spent on partner side +time spent on stranger side).

## Fluorescence in situ hybridization

Fluorescence in situ hybridization (FISH) was conducted using RNAscope Multiplex Fluorescent Reagent Kit v2 (Advanced Cell Diagnostics 323100) following the manufacturer's protocol. Voles were perfused transcardially with 0.1 M PBS (pH 7.4) followed by 4% paraformaldehyde in 0.1 M PBS. Then, brains were collected and postfixed for 24 hr in 4% paraformaldehyde at 4 °C followed by 24 hr in 20% sucrose and 24 hr in 30% sucrose, and were cut into 12 µm (6 series/brain) coronal sections using a cryostat (Leica Biosystems, Germany). Sections were mounted on slides and stored at –80 °C. Probes used in this study were RNAscope Prob-Mo-Ppib (#533491), RNAscope Negative Control Prob-DapB (#310043), RNAscope Probe-Mo-Drd1-C3 (#588161-C3), and RNAscope Probe-Mo-Drd2-C2 (#534471-C2). To verify the specificity and effectiveness of the rAAV-GCaMP6m virus, MSNs (co-labeled by D1R/D2R mRNA and D1-GCaMP6m/D2-GCaMP6m virus) were quantified. ×20 images including the NAc shell were acquired, and boxed areas (300×300 µm²) were selected. Three representative sections were chosen per brain. The NAc $^{D1/D2\ mRNA\ (AF594)}$ positive cells and co-labeled cells were then manually marked and counted using the 'multi-point' function of Image J (V1.8.0, National Institutes of Health, USA). The specificity ratio (%) was calculated as follows: (co-labeled cells of D1R or D2R mRNA and D1-GCaMP6m or D2-GCaMP6m) / (D1-GCaMP6m or D2-GCaMP6m positive cells). The effectiveness ratio (%) was calculated as follows: (co-labeled cells of D1R or D2R mRNA and D1-GCaMP6m or D2-GCaMP6m) / (D1R or D2R mRNA positive cells).

To validate D1R-mCherry or D2R-mCherry antibodies, FISH was conducted using the RNA-Protein Co-Detection Ancillary Kit (Advanced Cell Diagnostics 323180) following the manufacturer's protocol. Briefly, after pretreatment with the RNA-Protein Co-Detection Ancillary Kit, brain slices were incubated in primary antibody (1:300, ab183628, abcam, USA) at 4 °C overnight. Sections were then incubated with secondary antibody (anti-rabbit goat conjugated with Alexa Fluor 488, JacksonImmuno,

USA, 111-545-003, 1: 500) after hybridization with amplifiers. Sections were counterstained with DAPI (RNAscope Multiplex Fluorescent Reagent Kit v2, Advanced Cell Diagnostics 323100) for 30 s at room temperature. Glass slides were fixed with antifade solution, and coverslipped images were acquired under an Olympus microscope (OLYMPUS BX-43, OLYMPUS, Japan). To verify the specificity of the rAAV-mCherry virus, the MSNs co-labeled by D1R/D2R mRNA and D1-mCherry/D2-mCherry virus were quantified. ×20 images including the NAc shell were acquired, and boxed areas ($300 \times 300$ μm$^2$) were selected. Then, the D1-mCherry/D2-mCherry positive cells, NAc $^{D1/D2 \, mRNA \, (AF594)}$ positive cells, and co-labeled cells were manually marked. These positive or merged cells were counted using the 'multi-point' function of Image J. The numbers from three representative sections per brain were averaged as the value of each brain. The specificity ratio (%) was calculated as follows: (co-labeled cells of D1R or D2R mRNA and D1-mCherry or D2-mCherry positive cells) / (D1-mCherry or D2-mCherry positive cells).

## Statistical analysis

All data are represented as means ± SEMs. All data were assessed for normality using a one-sample Kolmogorov-Smirnov test, and the Levene's test was used to confirm homogeneity of variance. Comparisons between two groups were performed either by unpaired or paired t-tests. One-way analyses of variance (ANOVAs), two-way ANOVAs, or two-way repeated-measures ANOVAs were used to compare multiple groups under multiple testing conditions as appropriate. Post-hoc comparisons were conducted using Sidak. Kruskal-Wallis analyses were used to compare data when multiple groups of data did not meet the normality and homogeneity of variance requirements. Statistical procedures were performed using SPSS 20.0. All statistical graphs/charts were plotted via GraphPad Prism 6.0. All experiments and statistical analyses used the double-blind method. Significant levels were set at $p < 0.05$.

## Acknowledgements

This work was supported by STI2030-Major Projects (2022ZD0205101), the National Natural Science Foundation of China (31970424 and 31901082), the Natural Science Foundation of Shaanxi Province, China (2020JQ-412), the China Postdoctoral Science Foundation (2019M653534), the Department of Science and Technology of Shaanxi province (No. 2022PT-44), the Natural Science Basic Research Plan in Shaanxi Province of China (2023-JC-YB-207), and the Fundamental Research Funds for Central University of China (GK202301012). The contribution by LJY was also supported by an NIH grant.

## Additional information

### Funding

| Funder | Grant reference number | Author |
| --- | --- | --- |
| Ministry of Science and Technology of the People's Republic of China | 2022ZD0205101 | Fa-Dao Tai |
| National Natural Science Foundation of China | 31970424 | Fa-Dao Tai |
| National Natural Science Foundation of China | 31901082 | Zhixiong He |
| Natural Science Foundation of Shaanxi Province | 2020JQ-412 | Zhixiong He |
| China Postdoctoral Science Foundation | 2019M653534 | Zhixiong He |
| Shaanxi Provincial Science and Technology Department | No. 2022PT-44 | Ting Lian |

| Funder | Grant reference number | Author |
|---|---|---|
| Natural Science Basic Research Program of Shaanxi Province | 2023-JC-YB-207 | Hui Qiao |
| Fundamental Research Funds for the Central Universities | GK202301012 | Zhixiong He |

The funders had no role in study design, data collection and interpretation, or the decision to submit the work for publication.

## Author contributions

Lizi Zhang, Data curation, Formal analysis, Investigation, Methodology, Writing - original draft, Writing – review and editing; Yishan Qu, Data curation, Software, Investigation, Methodology; Larry J Young, Methodology, Writing – review and editing; Wenjuan Hou, Data curation, Investigation, Methodology; Limin Liu, Jing Liu, Yuqian Wang, Lu Li, Xing Guo, Yin Li, Caihong Huang, Zijian Lv, Yi-Tong Li, Hao Feng, Data curation, Investigation; Rui Jia, Resources, Data curation, Investigation, Methodology; Ting Lian, Resources, Funding acquisition, Investigation; Hui Qiao, Resources, Funding acquisition, Methodology; Zhixiong He, Data curation, Software, Formal analysis, Funding acquisition, Methodology, Writing – review and editing; Fa-Dao Tai, Conceptualization, Resources, Funding acquisition, Methodology, Project administration, Writing – review and editing

## Author ORCIDs

Yishan Qu ⓘ https://orcid.org/0000-0002-1849-6361
Larry J Young ⓘ https://orcid.org/0000-0003-2044-2239
Jing Liu ⓘ https://orcid.org/0000-0002-1139-4127
Hui Qiao ⓘ https://orcid.org/0000-0002-2476-4342
Fa-Dao Tai ⓘ https://orcid.org/0000-0002-6804-4179

## Ethics

All laboratory procedures were conducted in accordance with the Guidelines for the Care and Use of Laboratory Animals in China and the regulations of the Animal Care and Use Committee of Shaanxi Normal University. This study protocol was reviewed and approved by the Academic Committee of Shaanxi Normal University, Special Committee of Scientific Ethics, Approval No. 202112005.

Reviewer #2 (Public review): https://doi.org/10.7554/eLife.100292.3.sa1
Reviewer #3 (Public review): https://doi.org/10.7554/eLife.100292.3.sa2
Author response https://doi.org/10.7554/eLife.100292.3.sa3

# Additional files

## Supplementary files

MDAR checklist

## Data availability

All data generated or analysed during this study are included in the manuscript and source data files.

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
