## [Editor Report · eLife Assessment]

This **important** study advances our understanding of the role of dopamine in modulating pair bonding in mandarin voles by examining dopamine signaling within the nucleus accumbens across various social stimuli using state-of-the-art causal perturbations. The evidence supporting the findings is **compelling**, particularly cutting-edge approaches for measuring dopamine release as well as the activity of dopamine receptor populations during social bonding. Some concerns remain about the statistical analyses.

---

## [Referee Report · Reviewer #2 (Public review)]

Summary:

Using in vivo fiber-photometry the authors first establish that DA release when contacting their partner mouse increases with days of cohabitation while this increase is not observed when contacting a stranger mouse. Similar effects are found in D1-MSNs and D2-MSNs with the D1-MSN responses increasing and D2-MSN responses decreasing with days of cohabitation. They then use slice physiology to identify underlying plasticity/adaptation mechanisms that could contribute to the changes in D1/D2-MSN responses. Last, to address causality the authors use chemogenetic tools to selectively inhibit or activate NAc shell D1 or D2 neurons that project to the ventral pallidum. They found that D2 inhibition facilitates bond formation while D2 excitation inhibits bond formation. In contrast, both D1-MSN activation and inhibition inhibits bond formation.

Strengths:

The strength of the manuscript lies in combining in vivo physiology to demonstrate circuit engagement and chemogenetic manipulation studies to address circuit involvement in pair bond formation in a monogamous vole.

Weaknesses:

Weaknesses include that a large set of experiments within the manuscript are dependent on using short promoters for D1 and D2 receptors in viral vectors. As the authors acknowledge this approach can lead to ectopic expression and the presented immunohistochemistry supports this notion. It seems to me that the presented quantification underestimates the degree of ectopic expression that is observed by eye when looking at the presented immunohistochemistry. However, given that Cre transgenic animals are not available for Microtus mandarinus and given the distinct physiological and behavioral outcomes when imaging and manipulating both viral-targeted populations this concern is minor.

The slice physiology experiments provide some interesting outcomes but it is unclear how they can be linked to the in vivo physiological outcomes and some of the outcomes don't match intuitively (e.g. cohabitation enhances excitatory/inhibitory balance in D2-MSNs but the degree of contact-induced inhibition is enhanced in D2-MSN).

One interesting finding is that the relationship between D2-MSN and pair bond formation is quite clear (inhibition facilitates while excitation inhibits pair bond formation). In contrast, the role of D1-MSNs is more complicated since both excitation and inhibition disrupts pair bond formation. This is not convincingly discussed.

It seemed a missed opportunity that physiological read out is limited to males. I understand though that adding females may be beyond the scope of this manuscript.

Comments on revised version:

The authors addressed most of my comments, some would still need to be addressed.

(1) Previous comment: "The authors do not use an isosbestic control wavelength in photometry experiments, although they do use EGFP control mice which show no effects of these interventions, a within-subject control such as an isosbestic excitation wavelength could give more confidence in these data and rule out motion artefacts within subjects."

The authors should include a paragraph in the discussion addressing the limitations of not using an internal control for the fiberphotometric measurements.

(2) Previous Comment: The slice physiology experiments provide some interesting outcomes but it is unclear how they can be linked to the in vivo physiological outcomes and some of the outcomes don't match intuitively (e.g. cohabitation enhances excitatory/inhibitory balance in D2-MSNs but the degree of contact-induced inhibition is enhanced in D2-MSN).

My comment may not have been clear and the response didn't address my comment. What is missing in the discussion is an explanation of why a relative increase in excitation of D2-MSNs in the slice (Fig. 4J) is associated with an increased inhibition in vivo (Fig. 2H)?

(3) Previous Comment: One interesting finding is that the relationship between D2-MSN and pair bond formation is quite clear (inhibition facilitates while excitation inhibits pair bond formation). In contrast, the role of D1-MSNs is more complicated since both excitation and inhibition disrupt pair bond formation. This is not convincingly discussed.

Similarly, here the response provided does not address my question. Please focus on discussing why both excitation and inhibition of D1-MSNs can disrupt pair bond formation (Figure 7).

---

## [Referee Report · Reviewer #3 (Public review)]

Summary:

The manuscript is evaluating changes in dopamine signaling in the nucleus accumbens following pair bonding and exposure to various stimuli in mandarin voles. In addition, the authors present chemogenetic data which demonstrates excitation and inhibition of D1 and D2 MSN affect pair bond formation.

Strengths:

The experimental designs are strong. The approaches are innovative and use cutting-edge methods. The manuscript is well written.

Comments on revised version:

I appreciate the efforts by the authors to address many of my previous comments. The issues that remain are those associated with the statistics. It seems that not all statistical analyses were performed with the correct test. For example, the photometry data comparing emissions during partner vs stranger investigation over time would be best performed as a two-way ANOVA with odor type and time being separate variables. Also, there are paired t-tests being performed by calculating an average deltaF/F during the 4 second window following the being of a behavioral event. I think an area-under-the-curve calculation of these events would better capture the fluorescent emissions of these events as an index. Details in the Result describing the data being analyzed via ANOVA vs t-tests when reporting the results would be useful for the reviewer to understand each analysis.

---

## [Author Response]

The following is the authors’ response to the original reviews.

**Public Reviews:**

**Reviewer #1 (Public review):**
These experiments are some of the first to assess the role of dopamine release and the activity of D1 and D2 MSNs in pair bond formation in Mandarin voles. This is a novel and comprehensive study that presents exciting data about how the dopamine system is involved in pair bonding. The authors provide very detailed methods and clearly presented results. Here they show dopamine release in the NAc shell is enhanced when male voles encounter their pair bonded partner 7 days after cohabitation. In addition, D2 MSN activity decreases whereas D1 MSN activity increases when sniffing the pair-bonded partner.The authors do not provide justification for why they only use males in the current study, without discussing sex as a biological variable these data can only inform readers about one sex (which in pair-bonded animals by definition have 2 sexes). In addition, the authors do not use an isosbestic control wavelength in photometry experiments, although they do use EGFP control mice which show no effects of these interventions, a within-subject control such as an isosbestic excitation wavelength could give more confidence in these data and rule out motion artefacts within subjects.

We agree with your suggestion that mechanism underlying pair bonding in females should also be investigated. In general, natal philopatry among mammals is female biased in the wild(Greenwood, 1983; Brody and Armitage, 1985; Ims, 1990; Solomon and Jacquot, 2002); social mammals are rarely characterized by exclusively male natal philopatry (Solomon and Jacquot, 2002). Males often disperse from natal area to a new place. Thus, males rodents may play a dominant role in the formation and maintenance of mating relationships. This is a reason we investigate pair bonding in male firstly. Certainly, female mate selection, and sexual receptivity or refusal through olfactory cues from males, thereby affect the formation and maintenance of pair bonding (Hoglen and Manoli, 2022). This is also the reason why we should focus on the mechanisms underlying pair bonding formation in females in the future research. This has been added in the limitation in the discussion.

In photometry experiments, rAAV-D1/D2-GCaMP6m, a D1/D2 genetically encoded fluorescent calcium sensor, was injected into the NAc shell. The changes in fluorescence signals during these social interactions were collected and digitalized. To assess the specific response to social stimulus in fluorescence signals, changes in fluorescence signals during non-social behavioral bouts (such as freezing, exploration of the environment, grooming, rearing, etc…) were also recorded and analyzed. The result showed that dopamine release or D1/D2 MSNs activity displayed no significant changes after cohabitation of 3 or 7 days upon occurring of no-social behavior such as freezing, exploring, grooming and rearing. In addition, GCaMP6m is a genetically encoded calcium indicator. Changes in its fluorescence signal reflect changes in intracellular calcium ion concentration. Using EGFP virus as a control, it can be determined whether the fluorescence signal observed in the experiment is generated by the specific response of GCaMP6m to calcium or if there are other non-specific factors leading to fluorescence changes. If there is no similar fluorescence change in the EGFP control group, it can more strongly prove that the signal detected by GCaMP6m is a calcium-related specific signal. In some research article, they also use EGFP control group in photometry experiments (Yamaguchi et al., 2020; Qu et al., 2024; Zhan et al., 2024). Therefore, changes in fluorescence signals observed in the present study reflect neuron activities upon specific social behaviors, but were not affected by motion artefacts.

There is an existing literature (cited in this manuscript) from Aragona et al., (particularly Aragona et al., 2006) which has highlighted key differences in the roles of rostral versus caudal NAc shell dopamine in pair bond formation and maintenance. Specifically, they report that dopamine transmission promoting pair bonding only occurs in the rostral shell and not the caudal shell or core regions. Given that the authors have targeted more caudally a discussion of how these results fit with previous work and why there may be differences in these areas is warranted.

Thanks for your professional consideration. The brain coordinates of Bilateral 26-gauge guide cannulae were NAc (1.6 mm rostral, ± 1 mm bilateral, 4.5 mm ventral (for shell), 3.5 mm ventral (for core) from bregma) in report from Aragona et al (2006). In the present study, the brain coordinates of virus injection were (AP: +1.5, ML: ±0.99, DV: −4.2 (for NAc shell)). Thus, the virus injection sites were close to rostral shell in our study. However, as the diffusive expression of the virus, part of neurons in the rostrocaudal border and caudal shell also be infected by the virus, so we did not distinguish different subregions of NAc shell. In the future, we will use AAV13, a viral strategy could target / manipulate precise local neural populations, to address this issue. NAc is a complex brain structure with distinct regions that have different functions. Previous study suggested that GABAergic substrates of positive and negative types of motivated behavior in the nucleus accumbens shell are segregated along a rostrocaudal gradient (Reynolds and Berridge, 2001). However, a study found that food intake is significantly enhanced by administering μ-selective opioid agonists into the NAc, especially its shell region (Znamensky et al., 2001). Also, μ-opioid stimulation increases the motivation to eat (“wanting”) both in the NAc shell and throughout the entire NAc, as well as in several limbic or striatal structures beyond. For DAMGO stimulation of eating, the “wanting” substrates anatomically extend additionally beyond the rostrodorsal shell and throughout the entire shell (the caudal shell). Furthermore, DAMGO stimulates eating at NAc shell and core, as well as the neostriatum, amygdala…(Gosnell et al., 1986; Gosnell and Majchrzak, 1989; Peciña and Berridge, 2000; Zhang and Kelley, 2000; Echo et al., 2002; Peciña and Berridge, 2005, 2013; Castro and Berridge, 2014). In pair bond formation and maintenance, the rostral shell is the specific subregion of the NAc important for DA regulation of partner preference (Aragona et al., 2006). In conclusion, it appears that the changes in real time dopamine release and activities and electrophysiological properties of D1R, D2R MSNs in the NAc shell after pair bond formation may have primarily targeted to the rostral shell in our study, which is consistent with the report from Aragona et al.

The authors could discuss the differences between pair bond formation and pair bond maintenance more deeply.

Thanks for your suggestion. I have discussed the differences between pair bond formation and pair bond maintenance more deeply.

The dopamine and different types of dopamine receptors in the NAc may play different roles in regulation of pair bond formation and maintenance. The chemogenetic manipulation revealed that VP-projecting D2 MSNs are necessary and more important in pair bond formation compared to VPprojecting D1 MSNs. It is consistent with previous pharmacological experiments that blocking of D2R with its specific antagonist, while D1R was not blocked, can prevent the formation of a pair bond in prairie voles (Gingrich et al., 2000). This indicates that D2R is crucial for the initial formation of the pair bond. D2R is involved in the reward aspects related to mating. In female prairie voles, D2R in the NAc is important for partner preference formation. The activation of D2R may help to condition the brain to assign a positive valence to the partner's cues during mating, facilitating the development of a preference for a particular mate. In addition, the cohabitation caused the DA release, the high affinity Gi-coupled D2R was activated first, which inhibited D2 MSNs activity and promoted the pair bond formation. And then, after 7 days of cohabitation, the pair bonding was already established, the significantly increased release of dopamine significantly activated Gs-coupled D1R with the low affinity to dopamine, which increased D1 MSNs activity and maintained the formation of partner preference. While D1R is also present and involved in the overall process, its role in the initial formation of the pair bond is not as dominant as D2R (Aragona et al., 2006). However, it still participates in the neurobiological processes related to pair bond formation. For example, in male mandarin voles, after 7 days of cohabitation with females, D1R activity in the NAc shell was affected during pair bond formation. The extracellular DA concentration was higher when sniffing their partner compared to a stranger, and this increase in DA release led to an increase in D1R activity in the NAc shell. In prairie voles, dopamine D1 receptors seem to be essential for pair bond maintenance. Neonatal treatment with D1 agonists can impair partner preference formation later in life, suggesting an organizational role for D1 in maintaining the bond (Aragona et al., 2006). In pair-bonded male prairie voles, D1R is involved in inducing aggressive behavior toward strangers, which helps to maintain the pair bond by protecting it from potential rivals. In the NAc shell, D1 agonist decreases the latency to attack same-sex conspecifics, while D1 antagonism increases it (Aragona et al., 2006). In summary, D2R is more crucial for pair bond formation, being involved in reward association and necessary for the initial development of the pair bond. D1R, on the other hand, is more important for pair bond maintenance, being involved in aggression and mate guarding behaviors and having an organizational role in maintaining the pair bond over time. We therefore suggest that D2 MSNs are more predominantly involved in the formation of a pair bond compared with D1 MSNs.

The authors have successfully characterised the involvement of dopamine release, changes in D1 and D2 MSNs, and projections to the VP in pair bonding voles. Their conclusions are supported by their data and they make a number of very reasonable discussion points acknowledging various limitations
**Reviewer #2 (Public review):**
Summary:Using in vivo fiber-photometry the authors first establish that DA release when contacting their partner mouse increases with days of cohabitation while this increase is not observed when contacting a stranger mouse. Similar effects are found in D1-MSNs and D2-MSNs with the D1MSN responses increasing and D2-MSN responses decreasing with days of cohabitation. They then use slice physiology to identify underlying plasticity/adaptation mechanisms that could contribute to the changes in D1/D2-MSN responses. Last, to address causality the authors use chemogenetic tools to selectively inhibit or activate NAc shell D1 or D2 neurons that project to the ventral pallidum. They found that D2 inhibition facilitates bond formation while D2 excitation inhibits bond formation. In contrast, both D1-MSN activation and inhibition inhibit bond formation.Strengths:The strength of the manuscript lies in combining in vivo physiology to demonstrate circuit engagement and chemogenetic manipulation studies to address circuit involvement in pair bond formation in a monogamous vole.Weaknesses:Comment: Weaknesses include that a large set of experiments within the manuscript are dependent on using short promoters for D1 and D2 receptors in viral vectors. As the authors acknowledge this approach can lead to ectopic expression and the presented immunohistochemistry supports this notion. It seems to me that the presented quantification underestimates the degree of ectopic expression that is observed by eye when looking at the presented immunohistochemistry. However, given that Cre transgenic animals are not available for Microtus mandarinus and given the distinct physiological and behavioral outcomes when imaging and manipulating both viral-targeted populations this concern is minor.

Thanks for your professional comment. The virus used in the present study were purchased from brainVTA company. D1/D2 receptor promoter genes were predicted and amplified for validation by the company. The promoter gene was constructed and packaged by aav virus vector (taking rAAV-D2-mCherry-WPRE-bGH_polyA virus as an example, Author response image 1A). The D1/D2 promoter sequence is shown in the Author response image 1B-C. In addition, the D1 receptor gene promoter and D2 receptor gene promoter viruses used in this paper have been used in several published papers with high specificity (Zhao et al., 2019; Ying et al., 2022). In our paper, a high proportion of virus and mRNA co-localization was found through FISH verification and also showed high specificity of virus (Figure S15, S16).

**Author response image 1. sa3fig1:** (A) Gene carrier of rAAV-D2-mCherry-WPRE-bGH_polyA. (B-C) Gene sequence of D1 promoter and D2 promoter.

The slice physiology experiments provide some interesting outcomes but it is unclear how they can be linked to the in vivo physiological outcomes and some of the outcomes don't match intuitively (e.g. cohabitation enhances excitatory/inhibitory balance in D2-MSNs but the degree of contact-induced inhibition is enhanced in D2-MSN).

Thanks for your comment. The present study found that the frequencies of sEPSC and sIPSC were significantly enhanced after the formation of a pair bond in NAc shell D2 MSNs. The excitatory/inhibitory balance of D2 MSNs was enhanced after cohabitation.These results are not consistent with the findings from fiber photometry of calcium signals. One study showed that NAc D2 MSNs was linked to both ‘liking’ (food consumption) and ‘wanting’ (food approach) but with opposing actions; high D2 MSNs activity signaled ‘wanting’, and low D2 MSNs activity enhanced ‘liking’. D2 MSNs are faced with a tradeoff between increasing ‘wanting’ by being more active or allowing ‘liking’ by remaining silent (Guillaumin et al., 2023). Therefore, the increase in frequencies of sEPSC and sIPSC in D2 MSNs may reflect two processes, liking and wanting, respectively. We thought that hedonia and motivation might influence D2 MSNs activity differently during cohabitation and contribute to the processing of pair bond formation in a more dynamic and complex way than previously expected.

Moreover, the frequencies of sEPSC and sIPSC were significantly reduced in the NAc shell D1 MSNs after pair bonding, whereas the intrinsic excitability increased after cohabitation with females.

The bidirectional modifications (reduced synaptic inputs vs. increased excitability) observed in D1 MSNs might result from homeostatic regulation. The overall synaptic transmission may produce no net changes, given that reductions in both excitatory and inhibitory synaptic transmission of D1 MSNs were observed. Also, increases in the intrinsic excitability of D1 MSNs would result in an overall excitation gain on D1 MSNs.

One interesting finding is that the relationship between D2-MSN and pair bond formation is quite clear (inhibition facilitates while excitation inhibits pair bond formation). In contrast, the role of D1-MSNs is more complicated since both excitation and inhibition disrupt pair bond formation. This is not convincingly discussed.

Considering the reviewer’s suggestion, the discussion has been added in the revised manuscript.

In the present study, DREADDs approaches were used to inhibit or excite NAc MSNs to VP projection and it was found that D1 and D2 NAc MSNs projecting to VP play different roles in the formation of a pair bond. Chemogenetic inhibition of VP-projecting D2 MSNs promoted partner preference formation, while activation of VP-projecting D2 MSNs inhibited it (Figure 6). Chemogenetic activation of D2 MSNs produced the opposite effect of DA on the D2 MSNs on partner preference, while inhibition of these neurons produced the same effects of DA on D2 MSNs. DA binding with D2R is coupled with Gi and produces an inhibitory effect (Lobo and Nestler, 2011). It is generally assumed that activation of D2R produces aversive and negative reinforcement. These results were consistent with the reduced D2 MSNs activity upon sniffing their partner in the fiber photometry test and the increased frequency and amplitude of sIPSC in the present study. Our results also agree with other previous studies that chemogenetic inhibition of NAc D2 MSNs is sufficient to enhance reward-oriented motivation in a motivational task (Carvalho Poyraz et al., 2016; Gallo et al., 2018). Inhibition of D2 MSNs during self-administration enhanced response and motivation to obtain cocaine (Bock et al., 2013). This also suggests that the mechanism underlying attachment to a partner and drug addiction is similar.

Besides, in the present study, the formation of partner preference was inhibited after activation or inhibition of VP-projecting D1 MSNs, which is not consistent with conventional understanding of prairie vole behavior. Alternatively, DA binding with D1R is coupled with Gs and produces an excitatory effect (Lobo and Nestler, 2011), while activation of D1R produces reward and positive reinforcement (Hikida et al., 2010; Tai et al., 2012; Kwak and Jung, 2019). For example, activation of D1 MSNs enhances the cocaine-induced conditioned place preference (Lobo et al., 2010). In addition, D1R activation by DA promotes D1 MSNs activation, which promotes reinforcement. However, a recent study found that NAc-ventral mesencephalon D1 MSNs promote reward and positive reinforcement learning; in contrast, NAc-VP D1 MSNs led to aversion and negative reinforcement learning (Liu et al., 2022). It is consistent with our results that activation of NAc-VP D1 MSNs pathway reduced time spent side-by-side and impaired partner preference after 7 days of cohabitation. In contrast to inhibition of D2 MSNs, we found that inhibition of the D1 MSNs did not elicit corresponding increases in partner preference. One possible explanation is that almost all D1 MSNs projecting to the VTA/ substantia nigra (SN) send collaterals to the VP (Pardo-Garcia et al., 2019). For example, optogenetically stimulating VP axons may inadvertently cause effects in the VTA/SN through the antidromic activation of axon collaterals (Yizhar et al., 2011). Therefore, chemogenetic inhibition of D1 MSNs may also inhibit DA neurons in VTA, subsequently inhibiting the formation of a pair bond.

The dopamine and different types of dopamine receptors in the NAc may play different roles in regulation of pair bond formation and maintenance. The chemogenetic manipulation revealed that VP-projecting D2 MSNs are necessary and more important in pair bond formation compared to VPprojecting D1 MSNs. It is consistent with previous pharmacological experiments that blocking of D2R with its specific antagonist, while D1R was not blocked, can prevent the formation of a pair bond in prairie voles (Gingrich et al., 2000). This indicates that D2R is crucial for the initial formation of the pair bond. D2R is involved in the reward aspects related to mating. In female prairie voles, D2R in the NAc is important for partner preference formation. The activation of D2R may help to condition the brain to assign a positive valence to the partner's cues during mating, facilitating the development of a preference for a particular mate. In addition, the cohabitation caused the DA release, the high affinity Gi-coupled D2R was activated first, which inhibited D2 MSNs activity and promoted the pair bond formation. And then, after 7 days of cohabitation, the pair bonding was already established, the significantly increased release of dopamine significantly activated Gs-coupled D1R with the low affinity to dopamine, which increased D1 MSNs activity and maintained the formation of partner preference. While D1R is also present and involved in the overall process, its role in the initial formation of the pair bond is not as dominant as D2R (Aragona et al., 2006). However, it still participates in the neurobiological processes related to pair bond formation. For example, in male mandarin voles, after 7 days of cohabitation with females, D1R activity in the NAc shell was affected during pair bond formation. The extracellular DA concentration was higher when sniffing their partner compared to a stranger, and this increase in DA release led to an increase in D1R activity in the NAc shell. In prairie voles, dopamine D1 receptors seem to be essential for pair bond maintenance. Neonatal treatment with D1 agonists can impair partner preference formation later in life, suggesting an organizational role for D1 in maintaining the bond (Aragona et al., 2006). In pair-bonded male prairie voles, D1R is involved in inducing aggressive behavior toward strangers, which helps to maintain the pair bond by protecting it from potential rivals. In the NAc shell, D1 agonist decreases the latency to attack same-sex conspecifics, while D1 antagonism increases it (Aragona et al., 2006). In summary, D2R is more crucial for pair bond formation, being involved in reward association and necessary for the initial development of the bond. D1R, on the other hand, is more important for pair bond maintenance, being involved in aggression and mate guarding behaviors and having an organizational role in maintaining the bond over time. We therefore suggest that D2 MSNs are more predominantly involved in the formation of a pair bond compared with D1 MSNs.

It seemed a missed opportunity that physiological readout is limited to males. I understand though that adding females may be beyond the scope of this manuscript.

We gratefully appreciate for your valuable comment. The reviewer 1 also concerned this issue. We made a following response.

In general, natal philopatry among mammals is female biased in the wild(Greenwood, 1983; Brody and Armitage, 1985; Ims, 1990; Solomon and Jacquot, 2002); social mammals are rarely characterized by exclusively male natal philopatry (Solomon and Jacquot, 2002). Males often disperse from natal area to a new place. Thus, male rodents may play a dominant role in the formation and maintenance of mating relationships. This is a reason we investigate pair bonding in male firstly. Certainly, female mate selection, and sexual receptivity or refusal through olfactory cues from males, thereby affect the formation and maintenance of pair bonding (Hoglen and Manoli, 2022). This is also the reason why we should focus on the mechanisms underlying pair bonding formation in females in the future research. This has been added in the limitation in the discussion.

**Reviewer #3 (Public review):**
Summary:The manuscript is evaluating changes in dopamine signaling in the nucleus accumbens following pair bonding and exposure to various stimuli in mandarin voles. In addition, the authors present chemogenetic data that demonstrate excitation and inhibition of D1 and D2 MSN affect pair bond formation.Strengths:The experimental designs are strong. The approaches are innovative and use cutting-edge methods.The manuscript is well written.Weaknesses:The statistical results are not presented, and not all statistical analyses are appropriate.Additionally, some details of methods are absent.

As you suggested, we added the detailed information in the revised manuscript.

**Recommendations for the authors:**

**Reviewer #1 (Recommendations for the authors):**
(1) Remove references to 'extreme significance' - p is set as a threshold and the test is either significant or not.

Thanks for your suggestion. We have removed 'extreme significance' in the revised manuscript.

(2) The second half of the abstract is a little confusing the use of activation/inhibition makes it difficult to read and follow, this could be re-worded for clarity.

Sorry for the confusing. We reorganized the sentence as following.

In addition, chemogenetic inhibition of ventral pallidum-projecting D2 MSNs in the NAc shell enhanced pair bond formation, while chemogenetic activation of VP-projecting D2 MSNs in the NAc shell inhibited pair bond formation.

**Reviewer #2 (Recommendations for the authors):**
(1) In many instances repeated measures are presented from the same mice (e.g. Figures 1F, I; S1BC). Repeated measures for each mouse should be connected with a line in the figures. This will allow the reader to visually compare the repeated measures for each animal.

Thanks for your careful consideration. As reviewer suggested, the figures have been changed.

(2) It is unclear to me how the time point 0 for sniffing was determined. How is the time point 0 for side-by-side contact determined?

Sniffing is a behavior for olfactory investigation and defined as animals uses nose to inspect any portion of the stimulus mouse’s body, including the tail. The time point 0 for sniffing was the beginning of sniffing behavior occurs. The side-by-side behavior is defined as significant physical contact with a social object and huddle in a quiescent state. The time point 0 for side-byside behavior was the beginning of side-by-side behavior occurs.

(3) Figure 1-3: For the fiber photometry data 7 events (sniffs) are shown in the heat maps. Are these the first 7 sniffs? What went into the quantification? It seems that DA and D1/D2 responses are habituating. This could be analyzed and would need to be discussed.

In the heat maps (Figure 1-3), we showed the mean fluorescence signal changes of every subject (n = 7 voles) upon sniffing partner, stranger or an object in the experiment, but not the fluorescence signal changes of sniffing events in one vole. The quantification of changes in mean fluorescence signal of all subjects was showed in Figure 1F, 1I, Figure 2F, 2I, Figure 3F and 3I.

(4) Generally, it is very difficult to obtain cell type selectivity using short promoters in viruses (the authors acknowledge this). Which D1 and D2 promoter sequences were used for obtaining specificity? The degree of ectopic expression looks much higher than the quantification (e.g. in Fig. 3b, 6C, 7C, S14A, C). Is this due to thresholding?

The virus used in the present study were purchased from brainVTA company. D1/D2 receptor promoter genes were predicted and amplified for validation. The promoter gene was constructed and packaged by aav virus vector (taking rAAV-D2-mCherry-WPRE-bGH_polyA virus as an example, Author response image 1A). The D1/D2 promoter sequence is shown in the Author response image 1B-C. In addition, the D1 receptor gene promoter and D2 receptor gene promoter viruses used in this paper have been used in several published papers with high specificity (Zhao et al., 2019; Ying et al., 2022). In the Figure 6C, the first image is the merged fluorescence images that were taken under different fluorescence channels with the 20X objective. The second and the third images were taken under 40X objective from field of white box in the first image. The second and the third images were merged into fourth one. Due to the different exposure time and intensity, the fluorescence photo taken at 40X are clearer compared to image taken at the 20X. For example, in the Figure 6C, the labeled-cells were presented as following (Author response image 2). In our paper,virus infection and mRNA through FISH verification were co-localized in a high proportion displaying high specificity of virus (Figure S15, S16).Certainly, the number of positive neurons may be dependent on visuality (thresholding). Only visible cells were counted. The cell counting results at Author response image 2B and 2C are similar to the quantification in the Figure 6C.

**Author response image 2. sa3fig2:** (A) Immunohistological image showing co-localization of hM3Dq- mCherry-anti expression (green), D2R-mRNA (red), and DAPI (blue) in the NAc shell. Scale bar: 100 μm. (B) The cell counts and the determination of colocalization of the 20× immunohistochemistry images. The marked neurons were counted with white dots. (C) The cell counts and the determination of colocalization of the 40× immunohistochemistry images. The marked neurons were counted with white dots.

(5) Figure 6D/7D: the time scale seems to be off for both traces (40 seconds). For the hM3D Gq experiment, only one trace is shown. It would be more convincing to provide an input-output curve from several mice and to statistically compare the curves.

Response: Thanks for your careful consideration. As reviewer suggested, the figure of resting membrane potentials before and after drug CNO exposure from several voles was added in the revised manuscript.

(6) The presence of GIRK channels in MSNs has been a long debate and hM4D Gi activation may mostly act at the level of terminals by inhibiting neurotransmitter release. For demonstrating hyperpolarization of the soma showing the resting membrane potential before and after drug CNO exposure would be more convincing.

Thanks for your careful consideration. As reviewer suggested, the figure of resting membrane potential before and after drug CNO exposure was added in the revised manuscript.

(7) It is unclear to me how far the slice physiology informs the in vivo physiology (e.g. cohabitation enhances excitatory/inhibitory balance in D2-MSNs but the degree of contact-induced inhibition is enhanced in D2-MSN; D2-MSNs become less responsive to DA in the slice yet but at the time of enhanced DA release D2-MSN activity is also strongly reduced).

The present study found that the frequencies of sEPSC and sIPSC were significantly enhanced after the formation of a pair bond in NAc shell D2 MSNs. The excitatory/inhibitory balance of D2 MSNs was enhanced after cohabitation. These results are not consistent with the findings from fiber photometry of calcium signals. One study showed that NAc D2 MSNs was linked to both ‘liking’ (food consumption) and ‘wanting’ (food approach) but with opposing actions; high D2 MSNs activity signaled ‘wanting’, and low D2 MSNs activity enhanced ‘liking’. D2 MSNs are faced with a tradeoff between increasing ‘wanting’ by being more active or allowing ‘liking’ by remaining silent (Guillaumin et al., 2023). Therefore, the increase in frequencies of sEPSC and sIPSC in D2 MSNs may reflect two processes, liking and wanting, respectively. We thought that hedonia and motivation might different influence D2 MSNs activity during cohabitation and contribute to the processing of pair bond formation in a more dynamic and complex way than previously expected.

Moreover, the frequencies of sEPSC and sIPSC were significantly reduced in the NAc shell D1

MSNs after pair bonding, whereas the intrinsic excitability increased after cohabitation with females.

The bidirectional modifications (reduced synaptic inputs vs. increased excitability) observed in D1 MSNs might result from homeostatic regulation. The overall synaptic transmission may produce no net changes, given that reductions in both excitatory and inhibitory synaptic transmission of D1 MSNs were observed. Also, increases in the intrinsic excitability of D1 MSNs would result in an overall excitation gain on D1 MSNs.

(8) One interesting finding is that the relationship between D2-MSN and pair bond formation is quite clear (inhibition facilitates while excitation inhibits pair bond formation). In contrast, the role of D1-MSNs is more complicated since both excitation and inhibition disrupt pair bond formation.The discussion of this would benefit from another attempt.

As reviewer suggested, the discussion was added in the revised manuscript.

In the present study, DREADDs approaches were used to inhibit or excite NAc MSNs to VP projection and it was found that D1 and D2 NAc MSNs projecting to VP play different roles in the formation of a pair bond. Chemogenetic inhibition of VP-projecting D2 MSNs promoted partner preference formation, while activation of VP-projecting D2 MSNs inhibited it (Figure 6). Chemogenetic activation of D2 MSNs produced the opposite effect of DA on the D2 MSNs on partner preference, while inhibition of these neurons produced the same effects of DA on D2 MSNs. DA binding with D2R is coupled with Gi and produces an inhibitory effect (Lobo and Nestler, 2011). It is generally assumed that activation of D2R produces aversive and negative reinforcement. These results were consistent with the reduced D2 MSNs activity upon sniffing their partner in the fiber photometry test and the increased frequency and amplitude of sIPSC in the present study. Our results also agree with other previous studies, which showed that chemogenetic inhibition of NAc D2 MSNs is sufficient to enhance reward-oriented motivation in a motivational task (Carvalho Poyraz et al., 2016; Gallo et al., 2018). Inhibition of D2 MSNs during self-administration enhanced response and motivation to obtain cocaine (Bock et al., 2013). This also suggests that the mechanism underlying attachment to a partner and drug addiction is similar.

Besides, in the present study, the formation of partner preference was inhibited after activation or inhibition of VP-projecting D1 MSNs, which is not consistent with conventional understanding of prairie vole behavior. Alternatively, DA binding with D1R is coupled with Gs and produces an excitatory effect (Lobo and Nestler, 2011), while activation of D1R produces reward and positive reinforcement (Hikida et al., 2010; Tai et al., 2012; Kwak and Jung, 2019). For example, activation of D1 MSNs enhances the cocaine-induced conditioned place preference (Lobo et al., 2010). In addition, D1R activation by DA promotes D1 MSNs activation, which promotes reinforcement. However, a recent study found that NAc-ventral mesencephalon D1 MSNs promote reward and positive reinforcement learning; in contrast, NAc-VP D1 MSNs led to aversion and negative reinforcement learning (Liu et al., 2022). It is consistent with our results that activation of NAc-VP D1 MSNs pathway reduced time spent side-by-side and impaired partner preference after 7 days of cohabitation. In contrast to inhibition of D2 MSNs, we found that inhibition of the D1 MSNs did not elicit corresponding increases in partner preference. One possible explanation is that almost all D1 MSNs projecting to the VTA/ substantia nigra (SN) send collaterals to the VP (Pardo-Garcia et al., 2019). For example, optogenetically stimulating VP axons may inadvertently cause effects in the VTA/SN through the antidromic activation of axon collaterals (Yizhar et al., 2011). Therefore, chemogenetic inhibition of D1 MSNs may also inhibit DA neurons in VTA, subsequently inhibiting the formation of a pair bond.

The dopamine and different types of dopamine receptors in the NAc may play different roles in regulation of pair bond formation and maintenance. The chemogenetic manipulation revealed that VP-projecting D2 MSNs are necessary and more important in pair bond formation compared to VPprojecting D1 MSNs. It is consistent with previous pharmacological experiments that blocking of D2R with its specific antagonist, while D1R was not blocked, can prevent the formation of a pair bond in prairie voles (Gingrich et al., 2000). This indicates that D2R is crucial for the initial formation of the pair bond. D2R is involved in the reward aspects related to mating. In female prairie voles, D2R in the NAc is important for partner preference formation. The activation of D2R may help to condition the brain to assign a positive valence to the partner's cues during mating, facilitating the development of a preference for a particular mate. In addition, the cohabitation caused the DA release, the high affinity Gi-coupled D2R was activated first, which inhibited D2 MSNs activity and promoted the pair bond formation. And then, after 7 days of cohabitation, the pair bonding was already established, the significantly increased release of dopamine significantly activated Gs-coupled D1R with the low affinity to dopamine, which increased D1 MSNs activity and maintained the formation of partner preference. While D1R is also present and involved in the overall process, its role in the initial formation of the pair bond is not as dominant as D2R (Aragona et al., 2006). However, it still participates in the neurobiological processes related to pair bond formation. For example, in male mandarin voles, after 7 days of cohabitation with females, D1R activity in the NAc shell was affected during pair bond formation. The extracellular DA concentration was higher when sniffing their partner compared to a stranger, and this increase in DA release led to an increase in D1R activity in the NAc shell. In prairie voles, dopamine D1 receptors seem to be essential for pair bond maintenance. Neonatal treatment with D1 agonists can impair partner preference formation later in life, suggesting an organizational role for D1 in maintaining the bond (Aragona et al., 2006). In pair-bonded male prairie voles, D1R is involved in inducing aggressive behavior toward strangers, which helps to maintain the pair bond by protecting it from potential rivals. In the NAc shell, D1 agonist decreases the latency to attack same-sex conspecifics, while D1 antagonism increases it (Aragona et al., 2006). In summary, D2R is more crucial for pair bond formation, being involved in reward association and necessary for the initial development of the bond. D1R, on the other hand, is more important for pair bond maintenance, being involved in aggression and mate guarding behaviors and having an organizational role in maintaining the bond over time. We therefore suggest that D2 MSNs are more predominantly involved in the formation of a pair bond compared with D1 MSNs.

(9) For the chemogenetic inhibition/excitation experiment please specify the temporal relationship between CNO injection and the behavioral testing. Are the DREADDs activated during the preference testing or are we only looking at the consequences of DREADD activation during cohabitation? This would impact the interpretation of the results.

Considering the reviewer’s suggestion, we have clarified the time of CNO injection and the behavioral testing. In chemogenetic experiments, male voles were injected with CNO (1 mg/kg, i.p. injection) or saline once per day during 7-days cohabitation period. On day 3 and day 7 of cohabitation, the partner preference tests (3 h) were conducted after 3h of injection. Anton Pekcec (Jendryka et al., 2019) found that, in mice, after 60 min of CNO injection (i.p.), free CNO levels had dropped surprisingly sharply in CSF and cortex tissue, CNO could not be detected after 60 min. However, associated biological effects are reported to endure 6 - 24 h after CNO treatment (Farzi et al., 2018; Desloovere et al., 2019; Paretkar and Dimitrov, 2019). For example, René He et al. (Anacker et al., 2018) showed that chemogenetic inhibition of adult-born neurons in the vDG promotes susceptibility to social defeat stress by using of DREADDs for 10 days, whereas increasing neurogenesis confers resilience to chronic stress. Moreover, Ming-Ming Zhang et al. (Zhang et al., 2022) revealed that the selective activation or inhibition of the IC-BLA projection pathway strengthens or weakens the intensity of observational pain while the CNO (1 mg/kg) was i.p. injected into the infected mice on days 1, 3, 5, and 7 after virus expression. Furthermore, in study of James P Herman et al. (Nawreen et al., 2020) chronic inhibition of IL PV INs reduces passive and increases active coping behavior in FST. Therefore, we believe that 7-day CNO injections can produce chronic effects on MSNs and alters the formation of partner preferences.

(10) Discussion: "The observed increase in DA release resulted in suppression of D2 neurons in the NAc shell". "In contrast, the rise in DA release increases D1 activity selectively in response to their partner after extended cohabitation." These statements would need to be weakened as causality is not shown here.

Thanks for your rigorous consideration. We have reorganized the discussion in the revised manuscript.

“The observed increase in DA release resulted in alterations in activities of D2 and D1 neurons in the NAc shell selectively in response to their partner after extended cohabitation.”

(11) It would help if the order of supplementary figures would match their order of figures appearance in the result section.

Thanks for your suggestion. We reorganized the order of appearance in the revised manuscript.

(12) This may be beyond the focus of the study but it would be very interesting to know whether the physiological responses to partner contact are similarly observed in females.

Thanks for your concern. It is regretful that we did not observe physiological responses of female to partner contact. We predict the females may show the similar response patterns to their partner. In the future, we will supplement the research on the mechanism of partner preferences in female voles.

**Reviewer #3 (Recommendations for the authors):**
The manuscript is evaluating changes in dopamine signaling in the nucleus accumbens following pair bonding and exposure to various stimuli in mandarin voles. The manuscript is generally wellwritten. The experiment designs seem strong, although there are missing details to fully evaluate them. The statistics are not completed correctly, and the statistical values are not reported making them even harder to evaluate. There are a lot of potential strengths in this research. However, my review is limited because I am limited in how to evaluate data interpretation when statistical analyses are not clear. I provide details below.Major(1) Statistics should be provided in the Results section. It is not clear how to evaluate the authors' interpretations without presenting the statistical data. What stats are being reported about viral expression in cells on lines 192-194? What posthocs? There is only one condition, so I assume the statistic was a one-sample t-test. The authors should report the t-value, df, and p-value. No post-hoc is needed. There are many issues like this, which makes reviewing this manuscript very difficult. If the statistics were not conducted properly and reported clearly, I do not have confidence that I can evaluate the author's interpretation of the results.

Thanks for your suggestion. We report the t-value, df, and p-value in the Results section.

(2) Statistical tests should be labeled correctly. ANOVAs (found in figure caption) for Figure 1 data are not repeated measures. Rather, they are one-way ANOVA (with stimulus as a within-subject variable).

We used one-way ANOVA to analyze the changes in fluorescence signals in figure1-3. In the experiment, the changes in fluorescence signals of every subject were collected upon sniffing the partner, an unknown female, and an object. So, we used One-Way Repeated Measures ANOVA to analyze the data.

(3) The protocol for behavioral assessment and stimulus presentation during fiber photometry recording is not clear. For example, the authors mention on line 662 that voles ate carrots during some of the recording sessions, but nothing else is described about the recording session. What was the order of stimulus presentation? What was the object provided? Why is eating carrots analyzed separately from object, partner, and stranger exposure?

Response: Sorry for the confusing. The detailed description has been added. After 3 and 7 days of cohabitation, males were exposed to their partner or an unfamiliar female (each exposure lasted for 30 min) in random order in a clean social interaction cage. The changes in fluorescence signals during these social interactions with their partner, an unfamiliar vole of the opposite sex, or an object (Rubik's Cube) were collected and digitalized by CamFiberPhotometry software (ThinkerTech). To rule out that the difference in fluorescence signals was caused by the difference in virus expression at different time points, we used the same experimental strategy in new male mandarin voles and measured the fluorescence signal changes upon eating carrot after 3 and 7 days of cohabitation (The male mandarin voles were fasted for four hours before the test.). Since sniffing (object, partner, and stranger) and eating carrot were not tested in the same males, we analyzed sniffing and eating carrot separately.

(4) Supplement figures would be better as figures instead of tables. Many effects are hard to interpret.

As you suggested, we added the information of Supplement table1 in results.

(5) Citations should be included to note when pair bonding occurs in mandarin voles.

As you suggested, we added the citation in the revised manuscript.

Minor(1) Add a citation for the statement that married people live longer than unmarried people (Lines 51-52).

As you suggested, we added the citation in the revised manuscript.

(2) There is a table labeling viral vectors, but the table is not titled properly or referenced in the methods section.

Thanks for our careful checking. We reorganized the table title and the table was also cited in the revised manuscript.

(3) Sentences on lines 608-610 and 610-612 seem redundant.

This sentence was corrected.

(4) This is a rather subjective statement "Carrots are voles' favorite food."

We reorganized the sentence in the revised manuscript.

"Carrots are voles' daily food."

Anacker C, Luna VM, Stevens GS, Millette A, Shores R, Jimenez JC, Chen B, Hen R (2018) Hippocampal neurogenesis confers stress resilience by inhibiting the ventral dentate gyrus. Nature 559:98-102.

Aragona BJ, Liu Y, Yu YJ, Curtis JT, Detwiler JM, Insel TR, Wang Z (2006) Nucleus accumbens dopamine differentially mediates the formation and maintenance of monogamous pair bonds. Nature neuroscience 9:133-139.

Bock R, Shin JH, Kaplan AR, Dobi A, Markey E, Kramer PF, Gremel CM, Christensen CH, Adrover MF, Alvarez VA (2013) Strengthening the accumbal indirect pathway promotes resilience to compulsive cocaine use. Nature neuroscience 16:632-638.

Brody AK, Armitage KB (1985) The effects of adult removal on dispersal of yearling yellow-bellied marmots. Canadian Journal of Zoology 63:2560-2564.

Carvalho Poyraz F, Holzner E, Bailey MR, Meszaros J, Kenney L, Kheirbek MA, Balsam PD, Kellendonk C (2016) Decreasing Striatopallidal Pathway Function Enhances Motivation by Energizing the Initiation of Goal-Directed Action. The Journal of neuroscience : the official journal of the Society for Neuroscience 36:5988-6001.

Castro DC, Berridge KC (2014) Opioid hedonic hotspot in nucleus accumbens shell: mu, delta, and kappa maps for enhancement of sweetness "liking" and "wanting". The Journal of neuroscience : the official journal of the Society for Neuroscience 34:4239-4250.

Desloovere J, Boon P, Larsen LE, Merckx C, Goossens MG, Van den Haute C, Baekelandt V, De Bundel D, Carrette E, Delbeke J, Meurs A, Vonck K, Wadman W, Raedt R (2019) Longterm chemogenetic suppression of spontaneous seizures in a mouse model for temporal lobe epilepsy. Epilepsia 60:2314-2324.

Echo JA, Lamonte N, Ackerman TF, Bodnar RJ (2002) Alterations in food intake elicited by GABA and opioid agonists and antagonists administered into the ventral tegmental area region of rats. Physiology & behavior 76:107-116.

Farzi A, Lau J, Ip CK, Qi Y, Shi YC, Zhang L, Tasan R, Sperk G, Herzog H (2018) Arcuate nucleus and lateral hypothalamic CART neurons in the mouse brain exert opposing effects on energy expenditure. eLife 7.

Gallo EF, Meszaros J, Sherman JD, Chohan MO, Teboul E, Choi CS, Moore H, Javitch JA, Kellendonk C (2018) Accumbens dopamine D2 receptors increase motivation by decreasing inhibitory transmission to the ventral pallidum. Nature communications 9:1086.

Gingrich B, Liu Y, Cascio C, Wang Z, Insel TR (2000) Dopamine D2 receptors in the nucleus accumbens are important for social attachment in female prairie voles (Microtus ochrogaster). Behavioral neuroscience 114:173-183.

Gosnell BA, Majchrzak MJ (1989) Centrally administered opioid peptides stimulate saccharin intake in nondeprived rats. Pharmacology, biochemistry, and behavior 33:805-810.

Gosnell BA, Levine AS, Morley JE (1986) The stimulation of food intake by selective agonists of mu, kappa and delta opioid receptors. Life sciences 38:1081-1088.

Greenwood PJ (1983) Mating systems and the evolutionary consequences of dispersal. The ecology of animal movement:116-131.

Guillaumin MCC, Viskaitis P, Bracey E, Burdakov D, Peleg-Raibstein D (2023) Disentangling the role of NAc D1 and D2 cells in hedonic eating. Molecular psychiatry 28:3531-3547.

Hikida T, Kimura K, Wada N, Funabiki K, Nakanishi S (2010) Distinct roles of synaptic transmission in direct and indirect striatal pathways to reward and aversive behavior. Neuron 66:896907.

Hoglen NEG, Manoli DS (2022) Cupid's quiver: Integrating sensory cues in rodent mating systems. Frontiers in neural circuits 16:944895.

Ims RA (1990) Determinants of natal dispersal and space use in grey-sided voles, Clethrionomys rufocanus : a combined field and laboratory experiment. Oikos 57:106-113.

Jendryka M, Palchaudhuri M, Ursu D, van der Veen B, Liss B, Kätzel D, Nissen W, Pekcec A (2019) Pharmacokinetic and pharmacodynamic actions of clozapine-N-oxide, clozapine, and compound 21 in DREADD-based chemogenetics in mice. Scientific reports 9:4522.

Kwak S, Jung MW (2019) Distinct roles of striatal direct and indirect pathways in value-based decision making. eLife 8.

Liu Z, Le Q, Lv Y, Chen X, Cui J, Zhou Y, Cheng D, Ma C, Su X, Xiao L, Yang R, Zhang J, Ma L, Liu X (2022) A distinct D1-MSN subpopulation down-regulates dopamine to promote negative emotional state. Cell Res 32:139-156.

Lobo MK, Nestler EJ (2011) The striatal balancing act in drug addiction: distinct roles of direct and indirect pathway medium spiny neurons. Front Neuroanat 5:41.

Lobo MK, Covington HE, 3rd, Chaudhury D, Friedman AK, Sun H, Damez-Werno D, Dietz DM, Zaman S, Koo JW, Kennedy PJ, Mouzon E, Mogri M, Neve RL, Deisseroth K, Han MH, Nestler EJ (2010) Cell type-specific loss of BDNF signaling mimics optogenetic control of cocaine reward. Science (New York, NY) 330:385-390.

Nawreen N, Cotella EM, Morano R, Mahbod P, Dalal KS, Fitzgerald M, Martelle S, Packard BA, Franco-Villanueva A, Moloney RD, Herman JP (2020) Chemogenetic Inhibition of Infralimbic Prefrontal Cortex GABAergic Parvalbumin Interneurons Attenuates the Impact of Chronic Stress in Male Mice. eNeuro 7.

Pardo-Garcia TR, Garcia-Keller C, Penaloza T, Richie CT, Pickel J, Hope BT, Harvey BK, Kalivas PW, Heinsbroek JA (2019) Ventral Pallidum Is the Primary Target for Accumbens D1 Projections Driving Cocaine Seeking. The Journal of neuroscience : the official journal of the Society for Neuroscience 39:2041-2051.

Paretkar T, Dimitrov E (2019) Activation of enkephalinergic (Enk) interneurons in the central amygdala (CeA) buffers the behavioral effects of persistent pain. Neurobiology of disease 124:364-372.

Peciña S, Berridge KC (2000) Opioid site in nucleus accumbens shell mediates eating and hedonic 'liking' for food: map based on microinjection Fos plumes. Brain research 863:71-86.

Peciña S, Berridge KC (2005) Hedonic hot spot in nucleus accumbens shell: where do mu-opioids cause increased hedonic impact of sweetness? The Journal of neuroscience : the official journal of the Society for Neuroscience 25:11777-11786.

Peciña S, Berridge KC (2013) Dopamine or opioid stimulation of nucleus accumbens similarly amplify cue-triggered 'wanting' for reward: entire core and medial shell mapped as substrates for PIT enhancement. The European journal of neuroscience 37:1529-1540.

Qu Y, Zhang L, Hou W, Liu L, Liu J, Li L, Guo X, Li Y, Huang C, He Z, Tai F (2024) Distinct medial amygdala oxytocin receptor neurons projections respectively control consolation or aggression in male mandarin voles. Nature communications 15:8139.

Reynolds SM, Berridge KC (2001) Fear and feeding in the nucleus accumbens shell: rostrocaudal segregation of GABA-elicited defensive behavior versus eating behavior. The Journal of neuroscience : the official journal of the Society for Neuroscience 21:3261-3270.

Solomon NG, Jacquot JJ (2002) Characteristics of resident and wandering prairie voles, Microtus ochrogaster. Canadian Journal of Zoology 80:951-955.

Tai LH, Lee AM, Benavidez N, Bonci A, Wilbrecht L (2012) Transient stimulation of distinct subpopulations of striatal neurons mimics changes in action value. Nature neuroscience 15:1281-1289.

Yamaguchi T, Wei D, Song SC, Lim B, Tritsch NX, Lin D (2020) Posterior amygdala regulates sexual and aggressive behaviors in male mice. Nature neuroscience 23:1111-1124.

Ying L, Zhao J, Ye Y, Liu Y, Xiao B, Xue T, Zhu H, Wu Y, He J, Qin S, Jiang Y, Guo F, Zhang L, Liu N, Zhang L (2022) Regulation of Cdc42 signaling by the dopamine D2 receptor in a mouse model of Parkinson's disease. Aging cell 21:e13588.

Yizhar O, Fenno LE, Davidson TJ, Mogri M, Deisseroth K (2011) Optogenetics in neural systems. Neuron 71:9-34.

Zhan S, Qi Z, Cai F, Gao Z, Xie J, Hu J (2024) Oxytocin neurons mediate stress-induced social memory impairment. Current biology : CB 34:36-45.e34.

Zhang M, Kelley AE (2000) Enhanced intake of high-fat food following striatal mu-opioid stimulation: microinjection mapping and fos expression. Neuroscience 99:267-277.

Zhang MM et al. (2022) Glutamatergic synapses from the insular cortex to the basolateral amygdala encode observational pain. Neuron 110:1993-2008.e1996.

Zhao J, Ying L, Liu Y, Liu N, Tu G, Zhu M, Wu Y, Xiao B, Ye L, Li J, Guo F, Zhang L, Wang H, Zhang L (2019) Different roles of Rac1 in the acquisition and extinction of methamphetamineassociated contextual memory in the nucleus accumbens. Theranostics 9:7051-7071.

Znamensky V, Echo JA, Lamonte N, Christian G, Ragnauth A, Bodnar RJ (2001) gammaAminobutyric acid receptor subtype antagonists differentially alter opioid-induced feeding in the shell region of the nucleus accumbens in rats. Brain research 906:84-91.